# Canal blocking optimization in restoration of drained peatlands

Iñaki Urzainki[1,2], Ari Laurén[2], Marjo Palviainen[3], Kersti Haahti[1], Arif Budiman[4], Imam Basuki[4,5], Michael Netzer[4], and Hannu Hökkä[1]

[1]Natural Resources Institute Finland (Luke), Latokartanonkaari 9, FI-00790, Helsinki, Finland
[2]School of Forest Sciences, Faculty of Science and Forestry, University of Eastern Finland, Joensuu Campus, PO Box 111, (Yliopistokatu 7), FI-80101 Joensuu, Finland
[3]University of Helsinki, Department of Forest Sciences, PO-Box 27 00014, Helsinki, Finland
[4]Winrock International, 2121 Crystal Drive, Suite 500, Arlington, VA 22202, USA
[5]Center for International Forestry Research (CIFOR), Situ Gede, Sindang Barang, Bogor 16115, Indonesia

**Correspondence:** Iñaki Urzainki (inaki.urzainqui@luke.fi)

**Abstract.** Drained peatlands are one of the main sources of carbon dioxide ($CO_2$) emissions globally. Emission reduction and, more generally, ecosystem restoration can be enhanced by raising the water table using canal or drain blocks. When restoring large areas, the number of blocks becomes limited by the available resources, which raises the following question: in which exact positions should a given number of blocks be placed in order to maximize the water table rise throughout the area? There is neither a simple nor an analytic answer. The water table response is a complex phenomenon that depends on several factors, such as the topology of the canal network, site topography, peat hydraulic properties, vegetation characteristics and meteorological conditions. We developed a new method to position the canal blocks based on the combination of a hydrological model and heuristic optimization algorithms. We simulated three-day drydowns from a water saturated initial state for different block positions using the Boussinesq equation, and the block configurations maximizing water table rise were searched by means of Genetic Algorithm and Simulated Annealing. We applied this approach to a large drained peatland area (931 $km^2$) in Sumatra, Indonesia. Our solution consistently improved the performance of traditional block locating methods, indicating that drained peatland restoration can be made more effective at the same cost by selecting the positions of the blocks using the presented scheme.

## 1 Introduction

Peatlands occupy around 3% of global land area, but hold up to one third (630 Pg) of all carbon (C) held in active terrestrial pools (Page et al., 2011; Page and Baird, 2016; Xu et al., 2018; Quéré et al., 2018; Nichols and Peteet, 2019). In pristine conditions, peatlands typically act as C sinks, since the input of dead organic matter is usually greater than the biological decomposition of peat and other organic residues (Reddy and DaLaune, 2008). However, drainage may turn peatlands into C sources (Minkkinen and Laine, 1998; Hooijer et al., 2010; Ojanen et al., 2010; Jauhiainen et al., 2012). Drainage removes

excess water from peat and enhances site productivity, which is favourable for agriculture and forest production (Päivänen and Hånell, 2012; Evans et al., 2019). Even though drainage-based bioproduction can be economically viable, it has severe environmental drawbacks: it increases $CO_2$ emissions (Ojanen et al., 2010; Jauhiainen et al., 2012), the rate of peat subsidence (Couwenberg et al., 2010; Hooijer et al., 2010; Carlson et al., 2015; Evans et al., 2019), nutrient export to water courses (Nieminen et al., 2017) and fire risk in peatlands (Usup et al., 2004; Wösten et al., 2008; Page and Hooijer, 2016). $CO_2$ emissions have been particularly severe in managed tropical peatlands, where the annual $CO_2$ emission has been as high as 70 – 90 $Mg\ ha^{-1}$ (Hooijer et al., 2010; Jauhiainen et al., 2012). C emissions from tropical peatlands in Malaysia and Indonesia in 2015 corresponded to 1.6% of global fossil fuel emissions (Miettinen et al., 2017). According to Hooijer et al. (2010), the $CO_2$ emissions from drained peatlands in Indonesia range from 290 to 700 $Tg\ y^{-1}$.

Water table depth (WTD) has been found to be the key variable controlling $CO_2$ emissions from decomposition in tropical peatlands(Hooijer et al., 2010; Jauhiainen et al., 2012; Carlson et al., 2015; Evans et al., 2019). It has been estimated that raising the WTD from -80 to -40 cm would decrease $CO_2$ emissions on average by 50 $Mg\ ha^{-1}\ y^{-1}$ (Jauhiainen et al., 2012) and the rate of peat subsidence by 1.7 $cm\ y^{-1}$ (Evans et al., 2019). The reason behind the beneficial effects is that increasing water content in peat limits oxygen ($O_2$) supply for the decomposer organisms, and consequently slows down the rate of aerobic decomposition (Reddy and DaLaune, 2008). Therefore, raising the WTD is a powerful tool for peatland restoration, the aim of which is to establish a self-sustaining peat ecosystem that accumulates C.

Studies of canal and ditch blocking in temperate peatlands describe that WTD rise for peatland restoration has been commonly carried out using drain or canal blocks constructed from surrounding peat material, mineral soil or artificial materials (Ritzema et al., 2014; Armstrong et al., 2009; Parry et al., 2014). As discussed by Parry et al. (2014) the WTD response depends on site topography (Holden et al., 2006), block position (Holden, 2005), drain spacing and the hydraulic characteristics of peat (Dunn and Mackay, 1996). When restoring large peatland areas, the number of blocks becomes easily limited by available resources. This is especially important in tropical peatlands, where the canals are typically large, requiring big constructions that increase the cost of a single block (Armstrong et al., 2009; Ritzema et al., 2014). Working with limited resources raises a natural question: in which exact positions should a given amount of blocks be placed in order to maximize the amount of rewetted peat and consequently to minimize $CO_2$ emissions and the rate of subsidence?

To the best of our knowledge there is no systematic approach to support finding optimal block positions (Armstrong et al., 2009; Ritzema et al., 2014). Experimentally testing different block positions is impractical and inefficient. Process-based hydrological models, on the other hand, provide a useful tool to reveal changes in the WTD induced by different drainage setups (Dunn and Mackay, 1996). However, for large peatland areas and complex canal networks, process-based models on their own are not sufficient to solve for the best block positions, because the number of possible positions becomes subject to combinatorial explosion. To illustrate this, let us consider a setup with $b$ blocks having $n$ possible locations. The number of ways in which the blocks could be arranged equals $\binom{n}{b}$. For the case studied in this paper, $n = 11311$ and $b \approx 40$, and thus there are $\binom{11311}{40} = 1.6 \cdot 10^{114}$ ways to place them. Even with powerful computers it is not feasible to find the best combination by exhaustive search; a different strategy is required. By using global optimization methods such as Genetic Algorithm (GA) and Simulated Annealing (SA), it is possible to find approximate solutions to the problems without exhaustive search. Choos-

ing canal blocking positions is a combinatorial management problem for which global optimization methods are particularly suitable (Jin et al., 2016; Laurén et al., 2018; Rao, 2009).

Our objective in this work was to build a computational scheme based on a simple hydrological model coupled to an optimization algorithm that maximizes the amount of rewetted peat with a given number of canal blocks. The hydrological model uses the Boussinesq equation to compute WTD as a two-dimensional surface. Using the WTD -a proxy for the $CO_2$ emissions- as the target variable of the optimization problem, the optimization algorithms (GA and SA) look for the positions of the blocks that minimize the emissions. This scheme was applied to a drained peatland area ($931$ $km^2$) in Sumatra, Indonesia. Topographical details of the peatland areas, as well as rainfall data and physical peat properties were employed in the simulations. The implication of different canal blocking schemes will be discussed in the regional greenhouse gas emission context.

## 2   Material and Methods

### 2.1   Study area

The study area was located in Siak, Riau, Indonesia (Figure 1). The area belongs to humid tropical climate; the mean annual temperature is $27°C$ with very small monthly variation. The mean annual precipitation in the area is $2696$ $mm$, with the rainy season extending from October to April. The rainfall of the wettest month (November) exceeds $300$ $mm$ per month, while the driest month (July) receives $120$ $mm$ of rainfall. According to long-term weather statistics the mean dry period between the rainfall events during the dry season is 3.2 days and the maximum number of consecutive dry days is 20 (data from Pekanbaru Airport, located in the same province as the target area, years 1994-2013). Because of the humid climate and its topography, the area is characterized by tropical peatlands: the total area is $1100$ $km^2$, of which peatlands cover $931$ $km^2$. The depth of the peat deposit ranges from $2$ $m$ to $8$ $m$, the deepest peat deposit being located in the middle of the area, see Figure 2. Approximately 30% of the peat area represents hemic or moderately decomposed peat, and 60% is sapric or highly decomposed peat. The area was drained using canals of about 5m to $8$ $m$ metres wide, which are also used for transportation of wood and other products. The widest canals are captured in our dataset, but there exist smaller field drains that were omitted in this study due to the coarse resolution of the rasters. The total length of the canal network is $1100$ $km$. Typically, the canals are spaced in intervals of $500$ $m$ to $1000$ $m$.

For our computations we used the $100$ $m$ x $100$ $m$ resolution raster data shown in Figure 2, which together describe the surface elevation (DEM) (Vernimmen et al., 2019), the canal location and the peat depth and type. The DEM was preprocessed using the *fill sinks* algorithm in QGIS 3.4 in order to identify and fill unwanted surface depressions. The peat type influences the peat physical properties, $S_y$ and $T$, of the hydrological simulation, and the peat depth defines the impermeable bottom $ib$.

### 2.2   Computational scheme

The computation consists of the following modules: the canal water level subroutine, the hydrological model and an optimization algorithm. Figure 3 describes a single iteration in the optimization process. The canal water level subroutine computes

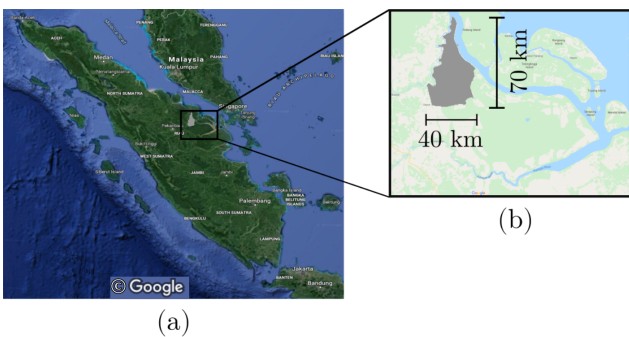

(a)

(b)

**Figure 1. (a)** Map of Sumatra Island, Indonesia, with the study area shown in grey. **(b)** Detailed view of the study area. Map data: © Google, Maxar Technologies.

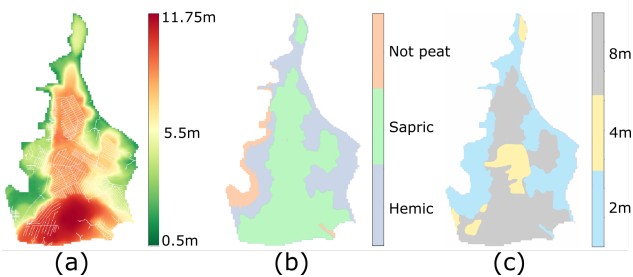

(a) (b) (c)

**Figure 2. (a)** DEM (coloured) with the canal network superposed (white), **(b)** peat types and **(c)** peat depth. The resolution of the rasters is 100 m x 100 m.

the canal water level (CWL) that would result from building canal blocks in some given positions. The CWL is passed to the peat hydrological model, which solves the WTD for the whole area, a quantity that is closely related to the target variable of the optimization problem, $\langle \zeta \rangle$, defined in section 2.2.2. The optimization algorithm evaluates the target variable and decides what canal block configuration to be studied next. This onsets a new iteration. We used Genetic Algorithm (GA) and Simulated Annealing (SA) optimization algorithms. We also tested an alternative, simpler optimization approach (SO) that maximizes the change in CWL instead (see Eq.(13)) and bypasses the hydrological simulation completely. See Table 1 for definitions of symbols used.

### 2.2.1 Canal water level subroutine

This subroutine calculates the CWL ($v'$) after building a set of blocks at positions $\boldsymbol{k}$, based on the original CWL ($\boldsymbol{v}$). In the absence of any blocks the CWL is assumed to be at a fixed distance, $w$, below the elevation derived from the DEM,

$$v_i = DEM_i - w. \tag{1}$$

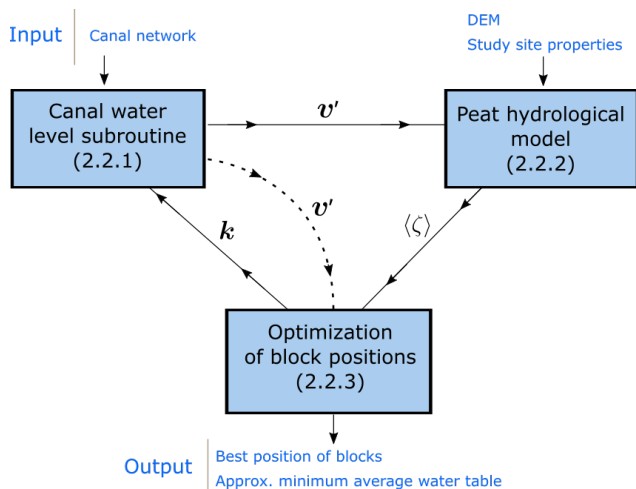

**Figure 3.** Schematic representation of a single iteration of the computation, showing the most relevant input and output and the interaction between the modules. The numbers in parentheses refer to the corresponding section in the main text. DEM stands for digital elevation model. The optimization algorithm proposes a particular position for the canal blocks, $\boldsymbol{k}$. Then, the canal water level subroutine computes the canal water level (CWL) resulting from that block placement, $\boldsymbol{v}'$. This information is passed on to the peat hydrological model, which solves for the WTD with $\boldsymbol{v}'$ as boundary conditions and computes the resulting target variable, the average WTD over 3 dry days, $\langle \zeta \rangle$, defined in 2.2.2. The optimization algorithm evaluates the performance and proposes a new $\boldsymbol{k}$ according to some rules specific to each algorithm. When using the alternative simple optimization strategy (SO), the CWL change, which depends only on $\boldsymbol{v}$ and $\boldsymbol{v}'$, see Eq.(13), is used as a target variable. This corresponds to the shortcut shown by the dashed arrows.

Here $i$ ranges over the set of pixels of the DEM that form the canal network, henceforth called *canal raster*. In our simulations, the value of $w$ was determined by direct observation on site and was set to $w = 1.2$m.

In order to compute how $\boldsymbol{v}$ would be affected by building a block in any pixel of the canal raster, information about the topology of the canal network is needed. In particular, it is necessary to know the direction of water flow to determine which adjacent pixels are upstream (and therefore potentially affected by the block). The direction of the water flow was inferred from the canal raster following two simple rules. For any two pixels in the canal network raster, we say that pixel B is a contiguous upstream pixel of A if and only if:

1. A and B are adjacent to each other (diagonal adjacency is also allowed).

2. A's water level is lower than B's, *i.e.*, $v_A < v_B$.

When a block is built in a given pixel of the canal raster its water level and the water level of upstream pixels rise up to match the height of the block with no delay. In what follows, instead of using the block height as a variable we use its complementary, the block head level $hl$. The block head level is defined as the distance from the DEM elevation to the highest point of the block

(Figure 4).

**Table 1.** Terms and symbols used in the study.

| Definition | Symbol | Units | Values/ref. |
|---|---|---|---|
| Simulated Annealing | SA | | |
| Genetic Algorithm | GA | | |
| Simple optimization | SO | | |
| Digital elevation model | DEM | | |
| Peatland area | $A$ | m$^2$ | $9.31 \cdot 10^8$ |
| Elevation of the peat surface | $s$ | m | from DEM |
| Canal water level measured from the sea level | CWL | m | |
| Vector representation of the CWL | $\boldsymbol{v}$ | | |
| CWL after building a set of blocks | $\boldsymbol{v}'$ | | |
| Number of pixels in the canal raster | $n$ | | 11311 |
| Canal block boolean vector | $\boldsymbol{k}$ | | Eq.(10) |
| Number of blocks | $b$ | | $0 \ldots 80$ |
| Block head level. Distance from peat surface to the highest point of the block | $hl$ | m | 0.2, 0.4 |
| Distance between DEM and CWL in the absence of any blocks | $w$ | m | 1.2 |
| Water table depth measured from the soil surface. Negative downwards. | WTD | m | |
| Spatial average of WTD | $\zeta$ | m | Eq.(5) |
| Temporal average of WTD over three days | $\langle \zeta \rangle$ | m | Eq.(7) |
| Hydraulic head | $h$ | m | |
| Precipitation | $P$ | mm d$^{-1}$ | 0 |
| Evapotranspiration | $ET$ | mm d$^{-1}$ | 3 |
| Impermeable bottom: depth of the peat deposit | $ib$ | m | from peat depth raster |
| Specific yield | $S_y$ | | |
| Hydraulic conductivity | $K$ | m d$^{-1}$ | |
| Transmissivity | $T$ | m$^2$ d$^{-1}$ | Eq.(4) |
| Marginal benefit | $MB$ | m$^3$ | Eq.(17) |

A detailed description of the algorithm used to implement these rules and compute $\boldsymbol{v}'$ is presented in Appendix A. The general response of the CWL to a block is schematically shown in Figure 4.

### 2.2.2 Peat hydrological model

The peat hydrological model simulates the two dimensional WTD surface for a given configuration of the blocks. From there it computes the target variable of the optimization algorithm, $\langle \zeta \rangle$, defined in Eq.(7). The WTD was solved using the Boussinesq equation, a quasi-3D groundwater flow partial differential equation (PDE) which is computationally much more efficient than

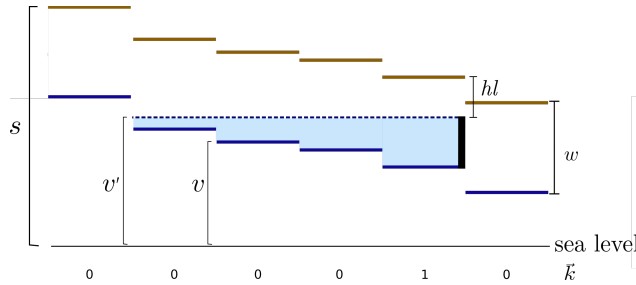

**Figure 4.** Side view of a canal. The blue and the brown horizontal solid lines represent the initial CWL, $v$, and the height of the peat surface, $s$, respectively. $w$ is a parameter that denotes the distance from the peat surface to the CWL. Each pixel is represented by one line segment. The vertical black line represents the block, and the dotted blue line represents the CWL after the block has been placed, $v'$. The shaded blue area represents the change in the CWL due to the block. The value of the vector $k$ is $k_i = 1$ if there is a block in pixel $i$ and otherwise $k_i = 0$.

solving the full 3D problem, and is a standard groundwater modeling equation for domains much wider than they are thick (Bear, 1979; Connorton, 1985; Skaggs, 1980; Koivusalo et al., 2000; Cobb et al., 2017),

$$S_y(h)\frac{\partial h}{\partial t} = \frac{\partial}{\partial x}\left(T(h)\frac{\partial h}{\partial x}\right) + \frac{\partial}{\partial y}\left(T(h)\frac{\partial h}{\partial y}\right) + P - ET, \tag{2}$$

where $S_y$ is the specific yield, $T$ is the transmissivity ($\mathrm{m^2\ d^{-1}}$), $h$ is the hydraulic head (m) and $P - ET$ is the difference between the precipitation and evapotranspiration ($\mathrm{m\ d^{-1}}$). The WTD is related to $h$ as follows,

$$WTD(x,y) = -\left[s(x,y) - h(x,y)\right], \tag{3}$$

where $s$ is the peat surface in meters above sea level. Equation (2) was numerically solved on a horizontal grid with a daily timestep using a finite volume solver (Guyer et al., 2009). Since Eq.(2) is a non-linear PDE, its solution at each timestep was
found iteratively so as to ensure numerical stability. The number of these internal iterations was set to three, which was regarded as a good compromise between accuracy and efficiency. The numerical scheme was fully implicit in time for $h$, and explicit for $T(h)$ and $S_y(h)$. The exterior faces of the grid were open water bodies, and Dirichlet -constant head- boundary conditions were applied on them. The value of $h$ at the canal pixels was forced to be equal to $v'$ by adding a source term large enough to completely dominate the corresponding term of the discretised equation (Versteeg and Malalasekera, 2007).
In this setup, the transmissivity is given by

$$T(h) = \int_{ib(x,y)}^{h(x,y)} K(x,y,z)dz, \tag{4}$$

where $K$ is the saturated hydraulic conductivity ($\mathrm{m\ d^{-1}}$) and $ib$ is the impermeable bottom. It follows from Eq.(4) that the transmissivity is a function of both $h$ and $ib$. However, since $ib$ has a known, fixed value for all the domain, we simplify the

notation by letting $T(h, ib) = T(h)$. The layered structure of the peat deposit, whose hydraulic conductivity $K(x, y, z)$ can vary in orders of magnitude along the vertical direction, $z$, is thus taken into account in $T(h)$. Since published hydraulic property profiles in tropical peat deposits are scarce (Baird et al., 2017), we parameterized the model based on the following:

- The degree of decomposition (hemic, sapric) affects the hydraulic conductivity. Hydraulic conductivity values for different decomposition stages were adopted from Wösten et al. (2008).

- Hydraulic conductivity decreases exponentially with depth (Koivusalo et al., 2000; Cobb et al., 2017).

- Woody peat is the dominant material in tropical peat deposits. The van Genuchten function was used to compute the volumetric water content of peat at depth $z$ for each degree of decomposition and $h$. In absence of measured tropical peat water retention characteristics, we used values from boreal woody peats with the same peat type and degree of decomposition (Päivänen, 1973). From the volumetric water content curves, the specific yield, $S_y$, the amount of water required for a differential increment in WTD elevation, was calculated.

Derived $T(h)$ and $S(h)$ curves for deepest substrate (10 m) hemic peat are shown in Figure 5(d).

Ponding water in fully saturated profiles was neglected and all surface water was removed from the computation, therefore assuming that the typical runoff velocity of water is greater than the infiltration velocity.

All simulations started from a fully saturated landscape, *i.e.*, $WTD = 0.0$m, or equivalently, $h = s$, which may occur after a heavy tropical rainfall event. Thereafter, for the optimization procedure, 3 dry days without any precipitation, $P = 0$mmd$^{-1}$ and $ET = 3$mmd$^{-1}$, were simulated with a daily timestep. The reason to adopt this particular setup is that the wet initial state acts as a system reset which, if followed by a period without precipitation, allows for qualitative comparison with observations. The exact number of dry days was decided according to two criteria. On the one hand, the mean of consecutive rainless days during the dry season in a 20 year time window was 3.2 days (data from Pekanbaru Airport, located in the same province as the target area, years 1994-2013). On the other hand, 3 timesteps results in a manageable computation load in the optimization process.

The spatially averaged WTD (m) at the end of each timestep $l$ was defined as

$$\zeta_l = \frac{1}{A} \iint WTD_l^*(x, y)\, dxdy, \tag{5}$$

where the integral extends to the whole peatland area including the canals, and $WTD_l^*$ stands for the solution of Eq.(2) at timestep $l$. The mean WTD over $d$ days is then given by

$$\langle \zeta \rangle_d = \frac{1}{d} \sum_{l=1}^{d} \zeta_l, \tag{6}$$

where the brackets $\langle \cdot \rangle$ denote temporal average. The average WTD over three days is specially relevant in this work, and in what follows we will denote it without subscripts,

$$\langle \zeta \rangle = \langle \zeta \rangle_3. \tag{7}$$

In order to estimate the annual $CO_2$ emissions that a given block configuration produces, the WTD for a full year was also
simulated. That simulation was also initialized with fully saturated initial conditions, and was made to coincide with a high rainfall event in December 2012. It was assumed that the yearly emitted amount of $CO_2$ per hectare, $m_{CO_2}$ (Mg ha$^{-1}$ y$^{-1}$) is proportional to $\langle \zeta \rangle_{365}$, *i.e.*,

$$m_{CO_2} = -\alpha \langle \zeta \rangle_{365} + \beta, \tag{8}$$

with coefficients (Jauhiainen et al., 2012)

$\alpha = 74.11$ Mg ha$^{-1}$m$^{-1}$y$^{-1}$
$\beta = 29.34$ Mg ha$^{-1}$y$^{-1}$. $\tag{9}$

The exact values of $\alpha$ and $\beta$ are important for the $CO_2$ emission estimation, but they are not relevant for the rest of the results produced in this work, since only the relative values of $m_{CO_2}$ are of interest in the optimization process. Instead, the crucial feature is that the annual average WTD is linearly related to the emitted amount of $CO_2$. The whole computational scheme is
therefore independent of the exact values of $\alpha$ and $\beta$, and they are only used at the last stage in order to report the results in units of annual emitted tonnes of $CO_2$.

### 2.2.3 Optimization of block positions

The management question of finding the position of a given number of blocks in such a way that the amount of emitted $CO_2$ or its proxy, $\langle \zeta \rangle$, is minimized can be formally formulated as follows.
Let $\boldsymbol{k} = (k_1, \ldots, k_n)$ be the boolean vector indicating presence or absence of a block in each canal pixel, *i.e.*,

$k_i = 1$ if there is a block in position $i$
$k_i = 0$ otherwise. $\tag{10}$

The objective function $f : \mathbb{R}^n \to \mathbb{R}$

$$f(\boldsymbol{k}) = \langle \zeta \rangle \tag{11}$$

maps a given block setup to $\langle \zeta \rangle$, the target variable. The objective function (or, equivalently, the target variable) is to be minimized subject to the constraint that

$$\sum_{i}^{n} k_i = b, \tag{12}$$

where $b$ is the number of blocks to be built. There is no analytic expression for $f$. Instead, it is a result of combining the canal blocking subroutine with the peat hydrological model. As pointed out in the Introduction, the search space is discrete

and too large for exhaustive search. Moreover, it might have many local minima that are not close to the global minimum, so algorithms that only seek local solutions are not useful. Therefore, this problem is better tractable with non-linear, global optimization algorithms.

    Even global optimization algorithms are not guaranteed to find the optimal solution in a convex search space in which all options cannot be tested. Given that there exists no guarantee that the process will converge towards the true global minimum of

$f$, the reliability of the optimization procedure benefits from exploring more than one optimization method. Genetic Algorithm (GA) and Simulated Annealing (SA) are heuristic methods that can often find the global minimum in many problems and are naturally applicable for the solution of discrete optimization tasks (Rao, 2009). In this case, both algorithms start off with some random $\boldsymbol{k}$ composed of $b$ blocks ($b = 0 \ldots 80$), for which the resulting $\langle \zeta \rangle$ is computed. Then, according to some rules specific to the algorithm, another $\boldsymbol{k}$ is proposed. This process is repeated for a fixed number of iterations, the same for all

numbers of blocks. Both algorithms tend to favor the configurations that result in a smaller value of the target variable $\langle \zeta \rangle$, but they also have the vital feature of avoiding getting stuck in local minima. In SA this is achieved by allowing disimprovements with certain probability. This probability is controlled by the sole parameter, the temperature (a term coming from metallurgy, where the inspiration for it came from), which decreases from an initial maximum value. In GA, on the other hand, the problem is circumvented by evaluating populations of *individual* vectors $\boldsymbol{k}$ at each iteration or generation. The fittest individuals are

passed on to the next iteration according to some rules that include mixing between individuals, also known as *mating*, and some randomness, or *mutations*. The mutation and the mating probabilities are the only parameters in the Genetic Algorithm implementation we used.

    The parameters used for both algorithms were fixed by trial and error, and they are shown in Table 2. The authors are aware that parallelizable versions of SA exist (see, *e.g.*, de Souza et al. (2010)), but the single processor algorithm was chosen for

this task. GA was run in parallel on 10 processors. With the same number of iterations (or generations), parallelization allows GA to explore 10 times more block configurations in a similar amount of time. SA was implemented by means of the Python package simanneal 0.5.0 (PyPi), and for GA the *eaSimple* algorithm in the DEAP library (Fortin et al., 2012) was used.

**Table 2.** Block locating methods and their parameters. The values of the parameters were decided empirically.

| Definition | SA | GA | SO | random | rule-based |
|---|---|---|---|---|---|
| Number of iterations or generations | 6000 | 6000 | 250000 | 2000 | manual |
| Number of processors | 1 | 10 | 10 | 1 | |
| Initial temperature | 300 | | | | |
| Final temperature | 1 | | | | |
| Single point crossover mating prob. | | 0.3 | 0.3 | | |
| Mutation probability | | 0.1 | 0.1 | | |

This optimization setup is computationally expensive, regardless of the optimization algorithm used. The main bottleneck of the computation is the numerical solution of the Boussinesq equation, Eq.(2). A simpler alternative is to maximize the CWL change,

$$\text{CWL change} = \sum_{i \in \text{canal raster}} (v'_i - v_i), \tag{13}$$

on its own. The CWL change is represented by the blue shaded area in Figure 4. The rationale behind this alternative choice of the target variable is simple: in general, it is to be expected that a higher CWL will lead to wetter peat throughout the area. By completely bypassing the numerical solution of the PDE, this approach would need a fraction of the computational resources required for the full optimization procedure described above, while potentially obtaining a good approximation of the minimum $\langle \zeta \rangle$. SO was implemented by modifying the target variable of GA and was run for 250000 iterations over 10 processors. This amounted to a similar computational effort as for the SA and GA algorithms.

To evaluate the performance of the optimization algorithms we compared the resulting $\langle \zeta \rangle$ against two other ways of positioning blocks: randomized and rule-based. The random block configurations were generated by randomly selecting locations from a uniform distribution. The value of $\langle \zeta \rangle$ from 2000 random block configurations was computed and aggregated into the mean, $\overline{\langle \zeta \rangle}_r$. The rule-based configuration was constructed following standard procedure in the absence of computational tools: blocks were placed in perpendicular intersections of contour line maps with the canal raster (Ritzema et al., 2014). The rule-based positions of the blocks for $b = 10$ are shown in Figure 7(a).

In order to enable a meaningful comparison between different setups, the average WTD resulting from these simulations was normalized with the average WTD in the absence of blocks. *i.e.*,

$$\langle \zeta^{(b)} \rangle_{norm} = \frac{\langle \zeta^{(b)} \rangle}{\langle \zeta^{(0)} \rangle}, \tag{14}$$

where $\langle \zeta^{(b)} \rangle$ is the $\langle \zeta \rangle$ resulting from placing $b$ blocks.

In a similar vein, we define the improvement of any block locating method to be

$$I^{(b)} = \langle \zeta^{(b)} \rangle - \langle \zeta^{(0)} \rangle. \tag{15}$$

It measures the simple difference in mean WTD between the reference value, $\langle \zeta^{(0)} \rangle$, and the one resulting from placing $b$ blocks with any of the methods above. In particular,

$$\overline{I}_r^{(b)} = \overline{\langle \zeta^{(b)} \rangle}_r - \langle \zeta^{(0)} \rangle \tag{16}$$

will be used to denote the mean improvement achieved by locating $b$ blocks randomly.

Yet some more insight can be gained by looking at the results in terms of marginal benefits. We define the marginal benefit of building $b + \Delta b$ blocks over $b$ blocks to be

$$MB(b) = \frac{\left| \langle \zeta^{(b+\Delta b)} \rangle_{norm} - \langle \zeta^{(b)} \rangle_{norm} \right|}{\Delta b}. \tag{17}$$

The quantities from Eqs.(14)–(17) are used to investigate the performance of all block placing methods in the task of minimizing $\langle \zeta \rangle$ with a fixed number of blocks.

## 3 Results

### 3.1 Reality check

In order to demonstrate that the peat hydrological model and the canal water level subroutine reproduce the expected qualitative behaviour of the WTD, two figures are shown. Figure 5 shows the WTD drop during three consecutive dry days for a cross section of the drained area. After three dry days, the WTD drops about 10 cm at the midpoint between two drains separated by 1.4 km. When the canals are closer to each other, WTD drop is larger, and if the canals are far apart enough the peat remains fully saturated. The shape of the WTD solution between two canals is the typical one for diffusion PDEs such as Eq.(2).

The behaviour of the canal water level subroutine is demonstrated by comparing the CWL change in a small drained area with and without canal blocks (Figure 6). The effect of the canal blocks on the CWL propagates to different distances depending on local topography. If the slope of $v$ is small, the effect of a single block can reach distances of the order of a kilometer. If, instead, $v$ changes very steeply, the effect of a block reaches less far. In addition, the amount of rewetted peat as a consequence of building one block is dependent on the local topography and physical properties of the peat deposit, and on the proximity to other canals. It is precisely the complexity of this response that calls for computational methods in order to solve for the optimal block placement.

### 3.2 Canal block optimization

The average WTD was computed using different scenarios with increasing number of canal blocks ($b = 5, \ldots, 80$) for each of the block placing methods described (rule-based, random, SA, GA, SO). Their resulting values are shown in Figure 7, and they constitute the main result of the present study.

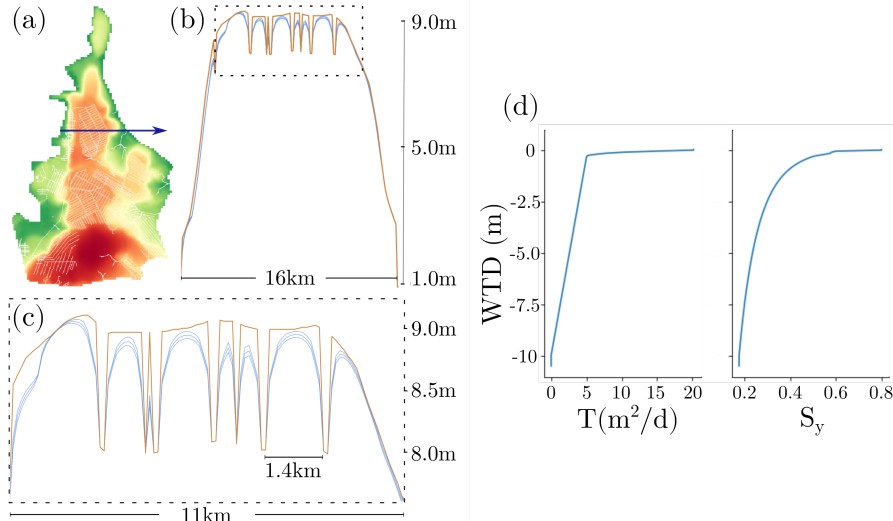

**Figure 5.** Cross section of the simulated WTD for three consecutive dry days after a big rainfall event, and peat hydraulic properties. **(a)** DEM (coloured) with the canal network superposed (white) and a straight horizontal line indicating the cross-section area (blue). **(b)** Peat surface (brown) and cross sectional view of the WTD (blue), measured in meters above sea level. The multiple blue lines correspond to the WTD for the three consecutive days of drydown. Abrupt low peat surface values correspond to canals. The dashed rectangle shows the region amplified in the figure below. **(c)** Magnified area from the figure above. **(d)** Transmissivity $T(h)$ and specific yield $S_y(h)$ functions for deepest substrate (10 m) hemic peat.

The most straightforward observation is that the more blocks there are the larger fraction of peat they will rewet, even if they are placed randomly. The second observation is that the optimization algorithms were able to find systematically better block positions than the random or the rule-based approaches. An informative way to gauge this difference is to realize that they were able to obtain with only 10 blocks the same amount of rewetted peat that the random configurations did with 60 blocks (Figure 7(b)). The largest performance difference of the optimization algorithms over the random happened for $b = 5$ and it was approximately $I_{GA}^{(5)} = 7 \cdot \overline{I}_r^{(5)}$ (Figure 7(c)). As the number of blocks increased, $I^{(b)}$ decreased monotonically for every block placement method. For the maximum amount of blocks considered, $b = 80$, $I_{SO}^{(80)} \approx 3 \cdot \overline{I}_r^{(80)}$. That is, at their best, the optimization algorithms were able to find block configurations that rewetted seven times more peat than the random and the rule-based approaches did for the same number of blocks; at their worst, they were three times better than the random.

Another thing to note is that the rate at which $\langle \zeta \rangle$ dropped for increasing $b$ was markedly slower for the random block placements than it was for the ones resulting from the optimization algorithms. This can be quantified by the marginal benefit , $MB(b)$ (Figure 7(d)), which gives the slope of Figure 7(b). For clarity, only the $MB$ for the best performing optimized solution is shown. $MB(b)$ for the mean of the random locations was approximately constant, while for the best optimized solution it decreased with $b$.

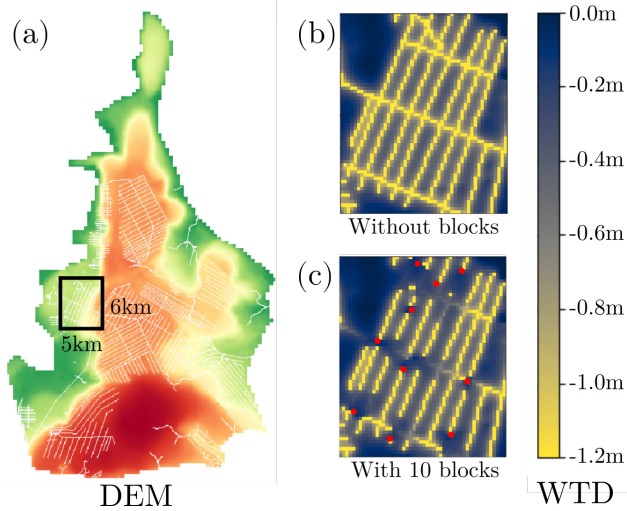

**Figure 6.** WTD after three dry days with and without blocks. **(a)** DEM (coloured) with the canal network superposed (white) and a rectangle indicating the area shown on the right. **(b)** WTD after three dry days without any blocks. **(c)** WTD after three dry days in the same area with ten blocks (block locations are indicated by red dots). WTD in the canal raster is defined as $v' - s$. Blocks help raise the WTD closer to the surface, but their effectiveness varies from each case depending on the local topography.

As Figure 7 shows, GA and SO performed similarly in the task of minimizing $\langle \zeta \rangle$. At first sight, this might look surprising, since the target variable for SO was not $\langle \zeta \rangle$ itself, but the CWL change. In order to understand this behaviour, we need to know how strongly $\langle \zeta \rangle$ and the CWL change are correlated with each other. Figure 8 shows that the optimal solutions for the two algorithms with $\langle \zeta \rangle$ as a target variable (SA and GA) tend to favour block configurations with smaller $\langle \zeta \rangle$, regardless of the CWL change, while SO is focused on maximizing CWL change, and gets a good performance in $\langle \zeta \rangle$ as a side product of the correlation between the two.

The sensitivity of $\langle \zeta \rangle$ to the block head level, $hl$, is demonstrated in Figure 9, where we plot $\langle \zeta \rangle_{norm}$ resulting from the best available block positions for two different values of the block head level, $hl = \{0.2\,\text{m}, 0.4\,\text{m}\}$. There can be a significant difference in the WTD, especially for large $b$.

### 3.3 Implication to CO$_2$ emissions

In order to draw further conclusions about the beneficial environmental impact of building canal blocks, we simulated the WTD for a full year under two different regimes: without any blocks and with the best available positions for the maximum number of blocks, 80. Rainfall intensity was taken from Pekanbaru Airport's weather station data, located in the same province as the target area. The big rainfall events registered during December 2012 were used as the starting point for the simulation, which was set up with completely saturated initial conditions. Evapotranspiration was set to $3\text{mm}\,\text{d}^{-1}$, and the block head level to $hl = 0.4\text{m}$. For each of the two block setups three daily WTD time series were recorded: the WTD in a drained area in

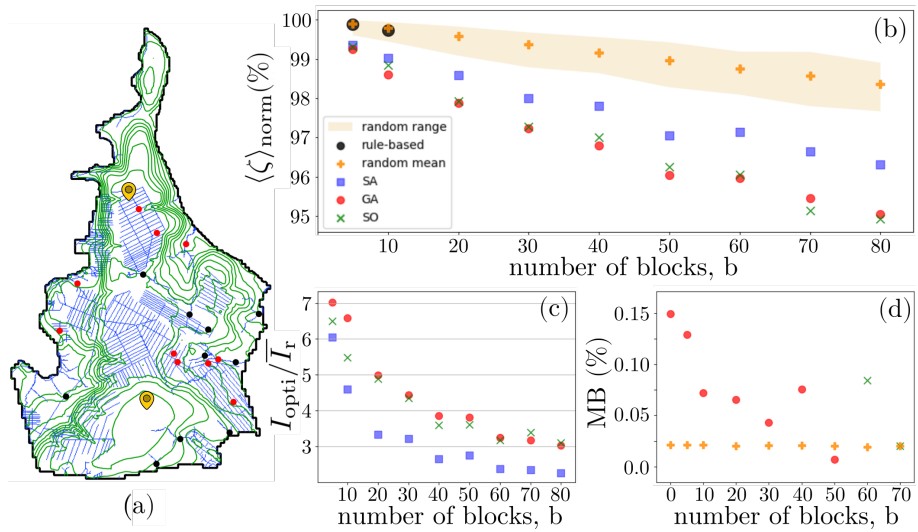

**Figure 7.** Peat rewetting performance comparison of random block locations, the rule-based approach and the optimization algorithms (SA: Simulated Annealing; GA: Genetic Algorithm; SO: simple algorithm) for different numbers of blocks. **(a)** Map of the area. The canal network is shown in blue, and the contour lines in green. The resulting block positions for the case $b = 10$, both for GA (red dots) and rule-based (black dots), are shown. Furthermore, the locations of the annual WTD simulations of Figure 10 are indicated by yellow plectrum-like markers.**(b)** $\langle \zeta \rangle_{norm}$, defined in Eq.(14), as a function of the number of blocks. The rule-based approach was only carried out for 5 and 10 blocks. **(c)** Relative improvement of several block locating methods with respect to the mean of the random, as defined in Eq.(16), for different numbers of blocks. **(d)** Marginal benefit, as defined in Eq.(17), for the best performing optimization algorithm and for the mean of the random configurations.

the north, the WTD in the natural undrained peat dome in the south (Figure 7 (a) shows the exact locations) and the spatially averaged WTD over the whole area, $\zeta$ (Figure 10).

Nearby blocks were able to raise the water table by approximately 20 cm in the chosen drained location. In the other end of the spectrum, the WTD in the natural zone was not affected at all. As a result, the effect of the 80 blocks in the WTD over the whole area, given by $\zeta$, was to raise it only by a few centimeters.

We obtained the following annual average values for the entire area: $\langle \zeta^{(0)} \rangle_{365} = -21.45$ cm without any blocks, and $\langle \zeta^{(80)} \rangle_{365} = -20.08$ cm, with the best available 80 blocks. In order to translate our results about the simulated annual WTD into the amount of emitted $CO_2$, we used Eq.(8). Thus, $m_{CO_2}^{(0)} = 45.34$ Mg ha$^{-1}$ y$^{-1}$ and $m_{CO_2}^{(80)} = 44.22$ Mg ha$^{-1}$ y$^{-1}$ were obtained for the aforementioned block configurations.

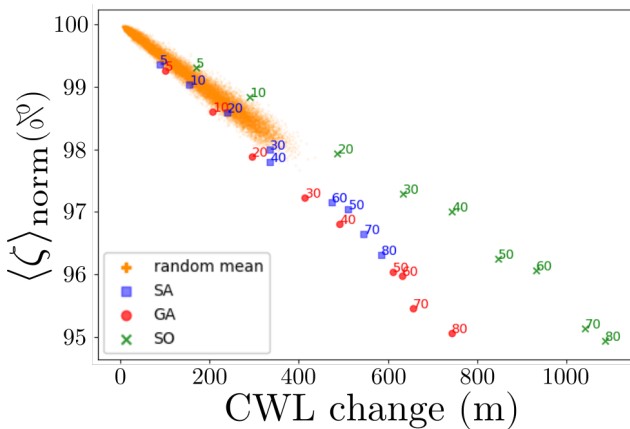

**Figure 8.** Correlation between $\langle \zeta \rangle$ and the CWL change for the random and the optimized block configurations. A larger block-induced change in CWL leads in general to a WTD closer to the surface. The number that accompanies each one of the points stands for $b$, the number of blocks that were located for each simulation.

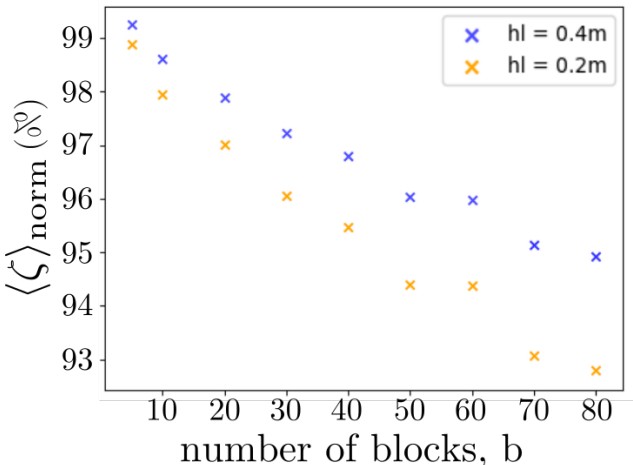

**Figure 9.** Sensitivity of the average WTD to a difference in the block head level, $hl$. The values of $\langle \zeta^{(b)} \rangle_{norm}$ correspond to the optimal block positions computed for $hl = 0.2$m (orange) and $hl = 0.4$m (blue). The larger the blocks are, the higher the WTD is risen.

## 4    Discussion

### 4.1    Model evaluation and reality check

To the best of our knowledge, this work introduces the first freely available systematic tool that can quantify the rewetting performance of different block configurations. It operates on all the easily available data (weather and GIS-derived data) and combines it in a scientifically coherent way. It is also designed to be computationally feasible for large areas. Therefore, this

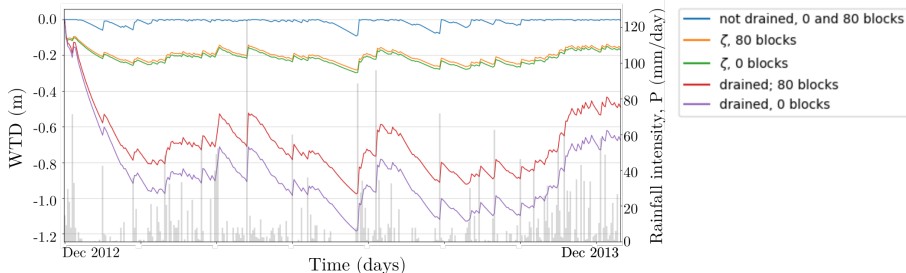

**Figure 10.** Simulated daily WTD for two sites (drained and natural, see Figure 7 for the exact locations) within the peatland area, and the average WTD, $\zeta$. The same period (December 2012 - December 2013) was simulated without any blocks (green and purple lines) and with the optimized 80 blocks (orange and red lines). The spatial average $\zeta$ for $b = \{0, 80\}$ is shown in orange and green. There was no appreciable difference in WTD in the undrained area between different block configurations, and the WTD is shown by a single line (blue line). Daily rainfall intensity is shown as grey vertical lines (data from Pekanbaru airport).

tool can potentially be very useful for decision makers in greenhouse gas emission mitigation and drained peatland restoration contexts.

The qualitative behaviour of the WTD and of the CWL in Figures 5 and 6 reflect the following expected traits. First of all, WTD decreases with time as a result of drainage. Second, the smaller the distance between canals, the more the WTD drops, for it was assumed that the system lacks any water input. In contrast, the WTD might stay close to the surface if the canals are far apart enough. Moreover, the effect of a set of blocks in the CWL propagates upstream in the correct way.

In this study we did not validate the hydrological model against actual field data, because there is no extensive, publicly available dataset. The aim of the paper was not to test a new hydrological model *per se*, but rather to solve a management question by applying a pre-existing one with parameter values derived from literature. We assume that a more precise parameterization would not have changed the outcome of the optimization procedure, and thus the qualitative assessment of the parameters' fitness was enough to fulfil our principal objective. It might be argued that in the absence of a quantitative validation, there is a high uncertainty in the simulated annual WTD of Figure 10. However, the simulated daily WTD of Figure 10 are in the same range and show similar dynamics as those reported earlier for drained peatlands in similar areas (Jauhiainen et al., 2012; Hooijer et al., 2012; Evans et al., 2019), and for natural peatland forests in Great Sunda islands (Cobb et al., 2017; Evans et al., 2019). Thus, we assume that WTD in Figure 10 and the consecutive $CO_2$ emissions, discussed in section 4.3, are plausible. Furthermore, we are aware that the hydrological model presented here may produce inaccurate estimates. The discretization error introduced with a daily timestep could be substantial, and the convergence test could be improved, for instance, by studying the behaviour of the solution with smaller time steps. However, accuracy and convergence needed to be sacrificed as a tradeoff against runtime. The hydrological model needed to be simplified just enough so that a meaningful amount of block setups could be explored and the management question could be successfully tackled..

Some remarks about the assumptions made in the canal water level subroutine are in order. As explained in section 2.2.1, the CWL in the absence of blocks was inferred from the DEM using a constant $w$ (see Eq.(1)). This implies that any local

fluctuation in the height of the DEM is directly transferred to the CWL. Indeed, a CWL derived in this manner is not expected to be monotonically decreasing in the direction of water flow. This non-monotonic nature of the CWL can lead to incorrect predictions of the effect a block has on the CWL. Another source of misrepresentation of the connectivity of the CWL comes from the artifact that the resolution of the DEM, 100 m x 100 m, introduces. According to the rules in section 2.2.1, if two different canals happen to be less than 100 m apart, then rule 1 will erroneously infer that those two pixels are in direct causal contact. Moreover, as mentioned in the description of the study area, there were small field drains that were not captured by the raster maps due to their coarse resolution. All these problems could be ameliorated by using a separate, complete canal network vector layer which contains the direction of the water flow. There is yet another class of approximations that were made in Eq.(1). First of all, in reality $w$ is not constant but it might vary in time due to seasonality, and in space at different heights. It is also worth noting that the resulting water profile after building a block is typically not a perfectly horizontal line, as depicted with dotted lines in Figure 4, but an inclined one. Furthermore, we are implicitly neglecting tidal effects, which could affect the water flow direction close to the seashore. All these approximations were either imposed by the quality of the data or judged to be of secondary importance in the computation of the CWL.

## 4.2 Canal block optimization

Two basic observations can be drawn from Figure 7. The first is that the performance of the rule-based approach is comparable to that of random location of the blocks. The positions for the blocks in the rule-based approach were located in perpendicular intersections between contour lines and canals (Ritzema et al., 2014), as shown in Figure 7(a). Figure 7(a) makes it apparent that it is very difficult to predict the effect of the blocks on the WTD by using logical reasoning alone: there are no evident differences between the locations of the blocks placed according to the rule-based and the GA methods. The rule-based approach was only carried out for 5 and 10 blocks, yet, as $b$ increases so does the complexity of the task, and it is therefore not expected that it would perform any differently from the random method when the amount of blocks increased. This leads us to conclude that the combination of the random trials and the rule-based approach may be interpreted as the best humanly possible results in the absence of any computational tools.

The second observation is that the optimization algorithms performed systematically better than the random and rule-based approaches. Going into further details, GA and SO were more successful in minimizing $\langle \zeta \rangle$ than SA. Under the same conditions, GA and SA are expected to perform similarly (Rao, 2009), but the single processor nature of SA restricted its search space to be 10 and 417 times smaller than those of GA and SO, respectively. The optimization performance of GA and SO was very similar for all numbers of blocks, but SO performed best for higher numbers of blocks. Both strategies are sound from the hydrological point of view, but their success in the optimization happens for different reasons. The good performance of SO can be explained by two factors. On the one hand, its simplicity allowed it to explore 42 times more block configurations than GA, thus being able to reach a fairly good approximation of the maximum CWL change even for large $b$. On the other hand, $\langle \zeta \rangle$ and the CWL change correlated strongly as is shown in Figure 8, meaning that SO got a good result in $\langle \zeta \rangle$ minimization as a byproduct of CWL change maximization. Another way of putting this is that, unlike the CWL change, $\langle \zeta \rangle$ gets the full 3D information about the catchment topography and the peat physical properties, but in return, the optimization task

is heavier. This may not be true for every study area. For instance, in domains with high spatial heterogeneity in peat physical properties the correlation is expected to be less evident. As the number of blocks to locate, $b$, increases, the size of the search space does so as $\binom{n}{b}$. It is this exponential increase in computational complexity what might explain the better performance of SO when the number of blocks is greater. Following this line of reasoning, the fact that SO performs better than GA only for $b = \{70, 80\}$ leads us to conclude that computational resources are limiting the performance of GA at least at those values of $b$, *i.e.*, a substantially better performance of GA is to be expected for high $b$ if the number of iterations increased. The success of both GA and SO calls for an alternative optimization strategy that would profit from both algorithm's strengths. Such an algorithm could be designed so that GA was initialized with several optima from the fast SO.

However interesting, comparing the performance of different algorithms was not the objective of this work. Instead, the main conclusion can be drawn by contrasting the outcome of the optimization algorithms with the best humanly available guesses. With the same number of blocks, the reduction in average WTD by the optimized block configuration is systematically greater than the one achieved simply by logical reasoning (Ritzema et al., 2014; Armstrong et al., 2009). This contrast is most significant for a small number of blocks, where the average WTD reduction resulting from the best available block locations is up to 7 times larger than the one derived from the mean of the random blocks (Figure 7(c)). As the number of blocks increases the relative improvement $I$ decreases, likely due to two main reasons. On the one hand, the aforementioned difficulty for the algorithms to find the optimal solution in an increasingly larger search space. On the other hand, the fact that the best positions might already be occupied by some of the blocks.

Another metric of interest to compare different block locating methods is the marginal benefit of adding one more block, shown in Figure 7(d). The marginal benefit for the random block configurations was almost constant, *i.e.*, the decrease of $\langle V_r^{(d)} \rangle$ was linear. This implies that if the blocks were to be built randomly, each additional block would be equally successful in reducing $\langle \zeta \rangle$. In contrast, the marginal benefit for the best available block locations varied with the number of blocks. Overall, it decreased as the number of blocks increased. This implies that the benefit of adding one more block decreases with the number of blocks that are already built. This fact is, once again, likely due to the two factors mentioned above. On the one hand, it is increasingly difficult for the algorithms to find an optimal solution in an exponentially increasing search space. On the other hand, for large number of blocks the most beneficial block locations are already occupied. Theoretically, there exists a limiting number of blocks at which the finite size of the area would make the marginal benefit to decrease even with the absolute best block locations. We suspect that with the current $b$ we were not yet at the limits of the system and that this finite-size phenomena will only be relevant for larger $b$.

It is not expected that a different choice of parameters would affect these general observations about the optimization results. While different parameterizations will result in a different WTD in absolute terms (see, *e.g.*, the case of varying $hl$, Figure 9), the relative differences in WTD between all block locating methods remain for different choices of parameter values.

It is also worth mentioning that solving the steady-state version of the Boussinesq equation, Eq. (2), was explored as the way to compute the target variable of the optimization, $\langle \zeta \rangle$. However, this approach was discarded in favour of the presented transient equation due to two observations. First, the steady-state solution does not yield a proper description of groundwater behaviour. In tropical climates rainfall is a key driver of hydrological processes and rainfall intensity is all but uniform in time.

Thus forcing the model to run with average rainfall and evapotranspiration does not result in a satisfactory model of these systems. Second, since the PDE is non-linear, the computational time needed to solve the steady-state version was comparable to the time needed to solve the transient equation.

## 4.3 Implication to $CO_2$ emissions

The simulated annual $CO_2$ emissions of section 3.3 are within the range of the values in the literature for peatlands in the same region (Hooijer et al., 2012; Evans et al., 2019). Relatively speaking, building 80 blocks to the whole 931 $km^2$ area mitigates only 2.24% of the $CO_2$ emissions. The reason for this modest performance might lie in 80 being too few blocks for such a large area. (Our method remains applicable for placement of a larger number of blocks, at the expense of longer computing times). Let us note that there are approximately 1100 $km$ of canals. When placing 80 blocks, the expected distance between a pair of blocks is about 14 $km$. Yet the influence a block has on the CWL spans, in our study area, a maximum of 2km. Let us stretch our results further to give a rough estimate of the number of blocks needed in order to prevent 10% of the emissions in the study area. Taking the values for 80 blocks as a reference, and assuming that $\langle \zeta^{(b)} \rangle$ decreases linearly with $b$, 350 blocks would be needed to reach that emission reduction goal. This would correspond to having on average one block every three kilometres. Of course, assuming that $\langle \zeta^{(b)} \rangle$ decreases linearly with $b$ is only a rough approximation (Figure 7 shows the true dependence). This nonlinear dependence points to the second reason for the modest performance of the 80 blocks: there seems to be room for improvement in our optimization procedure.

On the other hand, looking at the $CO_2$ emissions in absolute terms, building 80 blocks prevents the emission of 1.01 tonnes per hectare per year, or a total of 94156 tonnes annually throughout the whole area. To get a grasp of the magnitude of these numbers, they are of the order of what 25000 cars with an annual mileage of 20000 kilometres would emit.

It is clear that canal blocking raises WTD and therefore decreases $CO_2$ fluxes in tropical drained peatlands. The current application doesn't account for methane emissions, which might increase with rising WTD (Deshmukh et al., 2020; Manning et al., 2019). The optimization problem would have to be slightly reformulated to account for both negative and positive responses of C emissions to WTD rise. Yet the approach presented here would remain applicable provided that the hydrological model was extended to include a methane emission subroutine. This is left as a rather interesting open question for future work.

### 4.4 Application to real-life scenarios

When considering the applicability of our method to real-life scenarios, some of its underlying assumptions should be stated clearly. Our method assumes that it is possible to build a block at any given point in the canal raster and that the cost of doing so is constant and independent of site properties. Armstrong et al. (2009) carried out a comprehensive study of several drain blocking strategies in blanket peatlands in the UK. It is apparent from their work that the aforementioned assumptions do not hold in most real-life canal blocking scenarios. In particular, Armstrong et al. (2009) recommend building different types of blocks depending on the following site-specific variables: gradient of the CWL, canal width, peat wetness, peat depth, exposition of underlying mineral soil, and distance to building site. If our method is to have the desired practical impact, it should be able to accommodate these points. One way to do so would be to construct a realistic function that would return

block cost based on the above site properties. Indeed, the variables above may be easily translated into economical terms. For instance, a block built at a point of the CWL where the head gradient is large requires stable, expensive structures to avoid block failure. Similarly, a remote building site, wide canals and wet conditions increase the cost of building a block. Moreover, the bulk of the data needed to construct the block cost function is already part of the model (peat depth, DEM, WTD). Regarding the formulation of the optimization problem, block cost could be introduced by changing the constraint equation, Eq.(12): Instead of fixing the number of blocks, the block cost could be fixed.

It remains true that choosing the location of a set of blocks for best performance is a daunting task due to the complexity of the response of the water table, even more so when different types of blocks are considered. Therefore, the specifics of Figures 7 to 9 may change when several block types are considered, yet it is expected that the general trend would be similar: human guesses will not perform as well as optimized block locations. Nevertheless, the block-locating method described in this work will never replace expert knowledge. It should rather build upon it in order to have the desired practical impact. Our approach acknowledges that expert knowledge alone might not be enough to solve the rewetting problem of drained peatlands in an optimal way, and opens up the opportunity for local experts and organizations to use process-based hydrological modelling and numerical optimization techniques, which, as we have hopefully succeeded to show, can be powerful tools.

## 5  Conclusions

We constructed an optimization scheme that looks for the maximum water table rise for a drained peatland area given a fixed amount of canal blocks. Our results show that, with the same amount of resources (*i.e.*, number of blocks) the present computational setup enables a more effective canal blocking restoration of drained peatlands than human guesses do. The computational approach also enables cost-benefit analysis to solve several management questions.

*Code and data availability.*  The source code and the data used are available under the MIT license at https://github.com/LukeEcomod/blopti.

(Urzainki, 2020)

## Appendix A:  Canal water level subroutine

The information about the topology of the canal network was stored in a (sparse) matrix, $M$, of dimensions $(n \times n)$, where n is the number of pixels in the canal raster. For any two pixels of the canal raster, $i$ and $j$ , the entries of the matrix $M$ are

$M_{ij} = 1, \quad \text{if } j \text{ is a contiguous upstream pixel of } i$

$M_{ij} = 0 \quad \text{otherwise.}$ $\hspace{11cm}$ (A1)

Contiguous upstream pixels were defined in rules 1 and 2 of section 2.2.1. Note in particular that if $M_{AB} = 1$, that is, pixels A and B are adjacent and pixel B is upstream, it follows that $M_{BA} = 0$. Moreover, note that $M_{ii} = 0$ for any $i$. In other words,

$M$ is not symmetrical and all the elements of its diagonal are equal to zero. $M$ can then be interpreted as the adjacency matrix of the simple, directed graph $G$ whose nodes are the pixels of the canal raster and an edge exists if two nodes are in direct physical contact (Newman, 2018). In such a graph, the direction of the edges is the opposite to the direction of the water flow. Within this setup, the vector $\boldsymbol{k}' = \boldsymbol{k}M$, where $\boldsymbol{k}$ is the vector of the blocks' positions defined in Eq.(10), contains the information about all the first neighbours of the blocks in $\boldsymbol{k}$. Specifically,

$$k'_j = k_i M_{ij} = 1, \quad \text{if pixel } j \text{ is in direct causal contact with a block situated in pixel } i$$

$$k'_j = k_i M_{ij} = 0 \quad \text{otherwise.} \tag{A2}$$

Say we wish to build a block in pixel $A$, that is, $k_i = 1$ only for $i = A$. The operations that the canal water level subroutine performs in order to propagate the effect of this block to the neighbouring nodes of $A$ are described in Algorithm 1.

---

**Algorithm 1** Single iteration in the computation of $\boldsymbol{v}'$ from a $\boldsymbol{k}$ that consists of a single block in pixel $A$.

---

1: $v'_A \leftarrow s_A - hl$
2: $k' \leftarrow kM$
3: **for** $j$ in canal raster **do**
4:     **if** $k'_j = 1$ **and** $v_j < v'_A$ **then**
5:         $v'_j \leftarrow v'_A$.
6:     **end if**
7: **end for**

---

Line 1 sets the new value of the CWL in the pixel were the block is built to be $h$ units higher. In line 2 the neighbouring pixels that are in causal contact with pixel $A$ are stored into $k'$. The two conditions in line 4 effectively implement rules 1 and 2 of section section 2.2.1. Finally, for those pixels for which these two conditions are met, the CWL gets updated.

For the sake of readability, Algorithm 1 shows a single step in the process of computing $v'$, *i.e.*, it only updates the CWL for the first upstream pixels of a block located in A. In order to obtain the final CWL, the operations in Algorithm 1 would have to be iterated over for all successive $\boldsymbol{v}'$ until no more pixels were affected in the canal network. The algorithm could also be extended straightforwardly to any number of blocks. Following these rules, the CWL obtained after building a block looks like the one in Figure 4.

*Author contributions.* IU and AL contextualized the problem and developed the model code. IU performed the simulations. AB, IB and MN produced and validated the datasets. KH helped formulating research goals and methods. MP, HH and AL contributed with reviewing and editing the text. IU prepared the manuscript with contributions from all co-authors.

*Competing interests.* The authors declare that they have no conflict of interest

*Acknowledgements.* The authors wish to thank the referees' thorough comments on the manuscript, and Harri Koivusalo for useful discussions about the hydrological modeling. Furthermore, The authors wish to acknowledge CSC – IT Center for Science, Finland, for computational resources.

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
