# Peer review of "Canal blocking optimization in restoration of drained peatlands"

_Biogeosciences, 2020_

## Referee Comment (RC1) · Alex Cobb (Referee) · 15 Apr 2020

General comments:

This manuscript consists of a simple, quasi-hydrologic model for water level in ditches (involving a graph problem) plus a simple hydrologic model for flow in non-ditches. These models are used like a management tool to evaluate the impact of ditches on ground water levels and greenhouse gas emissions in the peatland. The scientific question is worthy. It is not obvious that it falls within the scope of BG; however, it does involve interaction between organisms and the geosphere in the sense that organic decomposition is responsible for the emissions that are inferred from the water table estimates. It seems to me that if Hooijer et al (2010; doi:10.5194/bg-7-1505-2010)

[Figure]

had a place in this journal, then so too does this manuscript. I have not seen such an exploration of peatland ditch blocking as an optimization problem in the literature. So the observations here, as well as some of the strategic decisions that the authors reached while designing their procedure, are novel, interesting and potentially useful.

In my view, the most valuable contributions of the paper are:

1. Showing that this problem is worth solving, in the sense that computer optimization of canal block locations worked much better than manual placement based on expert rules (though these rules are not sufficiently explained; see specific comments).

2. Showing that heuristic global optimization routines, in particular, can be valuable tools in attacking practical problems of this nature.

3. Suggesting that an initial optimization step of maximizing ditch water level could provide good starting steps for a more computationally expensive analysis of water levels within peat.

4. On a more technical note, the imposition of Dirichlet conditions in the domain interior using an implicit source term is a potentially useful approach for similar problems (though also not sufficiently explained; see specific comments).

The literature review is somewhat weak, in particular with reference to the tropical peat literature, and needs to make a clearer distinction between findings from higher-latitude and tropical peatlands. However, these additional references will not greatly change the narrative.

The results and the methods described appear valid and give no cause to suspect problems, and the simulation code has been made available (commendably). However, there is not enough detail in the manuscript to understand, even broadly, some aspects of what was done. I believe these clarifications can be made without adding supplementary material. See specific comments. The model for drydown in peat outside the canals is probably not very accurate (see specific comments), so accuracy for

this specific case is somewhat questionable, but this does not affect the main contributions noted above.

The manuscript is well structured overall. The abstract should broadly outline the methods that were used (simulated annealing, genetic algorithms, Boussinesq, three days' drydown after initial "reset"). The Methods and Discussion would benefit from a minor rearrangement of sections (see specific comments). The Discussion is reasonable, concise and avoids overreach.

There are three limitations that do not compromise the value of the manuscript but should be touched on in the Discussion and / or Introduction (for more on all of these, see specific comments):

1. Examining the area in Google Earth, it appears there are many field drains in rectangular arrays of about 60 m x 250 m that are disregarded in the simulations because of the grid resolution. Though this does not compromise the value of the paper, it does reduce the accuracy of the results and should be made clear to the reader in the Methods section and emphasized a bit more in the Discussion.

2. In practice, the expected head difference across a block is an important design criterion that was not considered in the optimization.

3. The effect of canal blocking on methane emissions should be part of an overall evaluation of impacts but relevant experimental data from tropical settings are lacking.

Specific comments:

- P1 L2, Abstract: "Ecosystem restoration can be achieved by raising the water table": "Achieved" is a rather strong word; rewetting seems to be a necessary but not sufficient condition for tropical peatland ecosystem restoration.

- Introduction: unclear to a reader which results have been described in the tropics. Please distinguish references from higher-latitude peatland studies; in particular, blanket peats are rather different systems from the lowland tropical peats examined here.

One approach could be to start by talking about peatlands in general, and then shift to discussing what is known from the tropics specifically.

- P1 L15: Additional references regarding peatland carbon pool: Nichols and Peteet 2019, Le Quéré et al 2018, or for a review, Page and Baird 2016.

- P2 L21: For peat subsidence in the tropics, see also: Couwenberg et al 2010, Hooijer et al 2010, Carlson et al 2015.

- P2 L22: Fire risk in peatlands: see also: Usup et al 2004, Page and Hooijer 2016.

- P2 L25: World Resources Institute: can you find a peer-reviewed (primary literature) source that makes this or a similar point?

- P2 L25-26: For $CO_2$ emissions from drained peatlands in Indonesia, consider also Miettinen et al 2017.

- P2 L27: "key variable controlling $CO_2$ emissions": add "from decomposition in tropical peatlands"

- P2 L27-28: By this point, it would be less confusing to focus on tropical references for $CO_2$ emissions vs water table depth; instead of Wilson et al 2011, consider Carlson et al 2015.

- P2 L33: Similarly, instead of Päivänen and Hånell 2012, consider Jauhiainen et al 2008, Ritzema et al 2014, Dohong et al 2018.

- P2 L35: Use tropical references again, then something like, "Studies of canal and ditch blocking in temperate peatlands have found that..."

- P2 L39: "This is especially important in tropical peatlands, where the canals are typically large:" Can a reference be provided? Armstrong et al (2009) describes the typical size of ditches in blanket bogs of the UK, and could be used as a point of contrast.

- P2 L50: Near: "Global optimization methods are commonly used...": Provide a very brief introduction to the terminology you use from optimization theory; I would guess that a majority of Biogeochemistry readers will not be able to infer what is "global" about global optimization, what is meant by "design space", nor why it is relevant that the design space is discontinuous and non-convex. It can be short.

- In the same place, you should very briefly introduce simulated annealing and genetic algorithms.

- Section 2, Materials and Methods: I suggest starting with the site description; reasons discussed further below.

- P3 L62: Can you come up with a different phrase or modifier for "the hydrological model", as the canal water level subroutine is also in some sense hydrological? Perhaps "peat hydrological model".

- P3 L66: "target variable": I believe this is the only place this phrase is used in the manuscript; consider changing to "objective function" to reduce the number of new terms for unfamiliar readers.

- P3 L67: "We also tested an alternative, simpler optimization approach (SO)": Simpler than what? Could be easier to follow if the SA and GA optimization approaches are introduced first.

- P3 L73: "This subroutine calculates the CWL after buiding a set of blocks, v', based on their positions, k." As written, it is not clear to what v refers (a set of blocks? blocks?), nor why it carries a prime ('). I suggest something like: "This subroutine calculates the CWL v' after building a set of blocks at positions k based on the original CWL v."

- P3 L74: "the CWL is assumed to be at a fixed distance, wd, below the peat surface, s...": Here, I ask myself: is wd a product? For this reason, I would discourage using compound symbols like this, but if they are used, please clarify in some way that this is a single symbol (perhaps by referring here to the nice table of symbols).

- P3 L75, Eqn 1: Does i index over pixels? if so, how is the peat surface elevation s defined in a canal pixel? It looks like the DTM pixels are much larger than the canals are wide, so I guess that s was derived directly from the DTM elevation? This would be easier to follow if the site (and DTM) were described first.

- P3 L76: "the value of wd was determined by direct observation...": Where? If at the site, it would simplify things to put the site description first. Otherwise, refer to that section.

- P3 L77: What is the "head level" of a block?

- P3 L78: Change "further up the canal network" to "upstream".

- P3 L77-81: Explain how "upstream" is determined prior to stating that a canal block causes the water in all upstream canal pixels to rise to the same level.

- P3-4 L77-85 and Appendix A: I think "direct causal contact" or "direct physical contact" does not convey what is meant here; it would be good to find a better phrase. How about talking about "contiguous upstream pixels", and explaining that "contiguous" means not separated by a canal block?

- P4 Figure 2: In this figure it becomes clear that v is positive up, and it appears that wd is positive down (if water table is further below surface s), but it is still not clear until Eqn 3 that WTD is positive up (even from Table 1). It would be good to mention this earlier, perhaps in Table 1, because "water table depth" causes different people to picture different things (does "a greater depth" mean the water table is higher or lower?). Throughout, I would suggest instead using "water level", after Bechtold et al (2014).

- P5 L90: Regarding the applicability of the Dupuit-Forchheimer assumptions: Insert "much", changing "for domains wider than they are thick" to "for domains much wider than they are thick".

- P5 Eqn 2: From Eqn 4, I believe that transmissivity T is a function of both the water

table h and the elevation of the impermeable bottom ib, so if the functional dependence of T is written, it should be T(h, ib) rather than T(h).

- P6 L96-97: How was time stepping handled? Explicit, implicit? How was convergence determined? From later in the page, it looks like time steps were fully explicit in the functions T and Sy (the value from the beginning of the time step was used)?

- P6 L99-100: "The value of h at the canal pixels was forced to be equal to v' by adding an implicit source term large enough to completely dominate the matrix diagonal": What was done exactly?

- P6 L110-114: Move the sentence "The van Genuchten function was used..." to before the sentence "In absence of measured..." (assuming that data from Päivänen 1973 were used to parameterize the van Genuchten function?)

- P6 L110-114: Plot the resulting specific yield and transmissivity functions. Transmissivity could be plotted for the lowest substrate elevation, for example (or curves with different substrate elevations could be plotted together).

- P6 L115: Were the values of T and Sy from the beginning of the time step used during time stepping? In any case, depending on the transmissivity function, I would guess the time discretization error with a daily time step could be substantial. But, the error could be acceptable as a tradeoff against runtime (at least when finding good candidate block positions). Convergence could be tested via multiple runs with decreasing time steps, but in my view is not strictly necessary for this paper.

- P6 L118-120: The broad outline of the simulation scenario (3 days of drydown from an initial "reset") are an important part of (SA and GA) objective function evaluation and should appear in the Abstract and the end of the Introduction.

- P7 L128: Does the spatial average of water table depth include canal pixels?

- P7 L128-133: I suggest dropping the subscript for the number of days averaged; it does not seem important for explaining the results and removing it would allow remov-

ing an equation (7).

- P7 Eqn 8: I am confused about sign issues in this equation (and / or Eqn 9). If I understand correctly, zeta is positive up, so this equation seems to suggest that emissions are larger if the mean water table is higher, which is surely not what is intended.

- Eqn 9: Are these parameter values, as well as the affine assumption (Eqn 8), from Jauhiainen et al 2012? If so, add a citation immediately after Eqn 9.

- P8 L149-150: "Let k = (k1, ..., kn) be the vector of block positions": How about: "Let k = (k1, ..., kn) be the boolean vector indicating presence or absence of a block in each canal pixel..."

- P8 L158-159: "design space", "search space": pick one, and use it also in the Introduction (P2 near L50).

- P8 L164: "Genetic algorithm... and simulated annealing... can find the global minimum with high probability": true for some problems, but it is not hard to cook one up for which they would not have odds that are good at all; for example, imagine an objective function that is 0 at a single point in the plane and 1 everywhere else. How about, "... can often find the global minimum in many problems."

- P9 Table 2: The empty column under "rule-based" is confusing. How about putting something here, like "Manual", or otherwise removing the column and simply describing it in the table caption.

- P9 L195-196: Regarding the rule-based configuration: What were the rules?

- P10 Eqns 15 and 16: It is not clear to me why an absolute value was used here. Why not just order the operands so that improvement in the mean WTD (higher water level) results in a positive value? Surely, if an intervention were to somehow lower the water table, this should result in a negative value (even if none of these interventions did so).

- P10, section 2.2, Study area: How was the DEM derived?

- P12 Figure 5: Nice figure. I guess the multiple water level lines are for the 3 consecutive days of drydown?

- P13 L248-249: "An informative way to gauge this difference...": where can I see this in the data?

- P14 Figure 7 caption: What is meant by "The random range was linearly interpolated"?

- P15 Figure 9: Improve caption; not obvious what is being shown without reference to the text.

- P16 L287: "this work introduces the first systematic tool..." It's hard to be sure; there could easily be something like this in use by large private landowners. How about "the first freely available tool" or "the first published study"?

- P17 L301: "the three-day average of the WTD would 2.13% units lower": I don't follow; where is this shown?

- P17 L306-: "Some remarks about the assumptions...": This would be a good place to mention any other limitations of the DEM. How was it derived? Do you believe it to be highly accurate?

- P17 Section 4.2: This section starts by talking about the rule-based method, then discusses the optimization results, and then comes back to compare the optimization results to the rule-based method. This section could be made easier to follow, and some redundancy might be eliminated, by starting with the optimization results and then contrasting the results from manual (and random) block placement.

- P17 L322: "The positions for the blocks in the rule-based approach were based on the contour map..." Derived how? Besides explaining this in the Methods, it would be good to briefly mention the criteria for manual block placement here.

- P19 L370-379: The discussion of why the idea of steady-state Boussinesq solution

was rejected could be condensed.

- P19 Section 4.3: In this section, it would be good to at least briefly mention the possible effects of canal blocks on methane emissions from ditches. From higher-latitude peatlands there are a number of studies, in Finland and elsewhere, e.g., Minkkinen and Laine (2006). In the tropics much less work has been done; without doing a search, I am aware of these two: Jauhiainen and Silvennoinen (2012), Manning et al (2019).

- P19 L384-385: 80 blocks results in about 14 km between blocks: you could state more explicitly that your method remains applicable for placement of a larger number of blocks (at the expense of more computing time). This could be a good place to mention the typical design criterion, not considered in this study, of avoiding large head differences across blocks to prevent dam failure (per general comments).

- P19 L392-394: Good point but phrasing could be improved.

- P20 Algorithm 1: v'A <- vA + hl: Appears not to match Figure 2, or Table 1; looks like it should be v'A <- sA - hl.

Technical corrections:

- P1 L4: Change "water level raise" to "water level rise".

- P3 L77: Change "its water level raises up" to "its water level rises up".

- P4 Figure 1: The references to text sections look wrong (assuming that is what they are?): I guess 2.2 in "Canal water level subroutine (2.2)" should be changed to 2.1.1, etc.

- P8 L170: Change "the only parameter" to "the sole parameter".

- P8 L172: Add quotes around "individual", because it is being used in the GA sense of an "organism". Change 'individual k' to '"individual" vectors k'.

- P8 L174: "our implementation": the analysis used an existing implementation of the

algorithm, no? Perhaps, "... were the only parameters in the genetic algorithm implementation we used."

- P9 L177: Change "wide spread classical single processor algorithm" to "single processor algorithm".

- P9 L188: Change "over 10 processors" to "on 10 processors".

- P10 L222: Change "sappric" to "sapric".

- P11 Figure 4: Colors for "2 m" and "8 m" bins are hard to distinguish; why not use another continuous scale?

- P13 L262: Remove "eloquently".

- P14 Figure 7: What are the plectrum-like markers?

- P14 Figure 7: How do I find the "natural" and "drained" areas discussed later in the text?

- P14 Figure 7: In the plots, make the markers bigger. They are hard to distinguish, and it is hard to make out their colours and shapes.

- P15 Figure 8: Move labels away from markers to make them easier to read.

- P15 Figure 8: Make it more obvious to the reader what is better and what is worse on the vertical axis.

- P15 Figure 8: CWL change (m) ranges to 1000; unit error? Or is this a sum?

- P16 Figure 10: Text for legend is very small; move outside axes? Or, consider labeling the lines directly; it takes some work, with reference to the caption, to figure out which line is which.

- P16 292: Change "lowers" to "decreases".

- P18 L340: Change "any" to "every".

- P20 L418: Change lh to hl.

References:

1. Bechtold, M., Tiemeyer, B., Laggner, A., Leppelt, T., Frahm, E., & Belting, S. (2014). Large-scale regionalization of water table depth in peatlands optimized for greenhouse gas emission upscaling. Hydrology and Earth System Sciences, 18(9), 3319–3339. doi:10.5194/hess-18-3319-2014

2. Carlson, K. M., Goodman, L. K., & May-Tobin, C. C. (2015). Modeling relationships between water table depth and peat soil carbon loss in Southeast Asian plantations. Environmental Research Letters, 10(7), 074006. doi:10.1088/1748-9326/10/7/074006

3. Couwenberg, J., Dommain, R., & Joosten, H. (2009). Greenhouse gas fluxes from tropical peatlands in south-east Asia. Global Change Biology, 16(6), 1715–1732. doi:10.1111/j.1365-2486.2009.02016.x

4. Hooijer, A., Page, S., Canadell, J. G., Silvius, M., Kwadijk, J., Wösten, H., & Jauhiainen, J. (2010). Current and future CO2 emissions from drained peatlands in Southeast Asia. Biogeosciences, 7(5), 1505–1514. doi:10.5194/bg-7-1505-2010

5. Jauhiainen, J., & Silvennoinen, H. (2012). Diffusion GHG fluxes at tropical peatland drainage canal water surfaces (Kasvihuonekaasujen diffuusio kuivatuskanavissa trooppisilla soilla). Suo 63(3-4): 93–105.

6. Le Quéré, C., Andrew, R. M., Friedlingstein, P., Sitch, S., Hauck, J., Pongratz, J., ... Canadell, J. G. (2018). Global Carbon Budget 2018. Earth System Science Data, 10(4), 2141–2194. doi:10.5194/essd-10-2141-2018

7. Manning, F. C., Kho, L. K., Hill, T. C., Cornulier, T., & Teh, Y. A. (2019). Carbon emissions from oil palm plantations on peat soil. Frontiers in Forests and Global Change, 2. doi:10.3389/ffgc.2019.00037

8. Miettinen, J., Hooijer, A., Vernimmen, R., Liew, S. C., & Page, S. E. (2017). From

carbon sink to carbon source: extensive peat oxidation in insular Southeast Asia since 1990. Environmental Research Letters, 12(2), 024014. doi:10.1088/1748-9326/aa5b6f

9. Minkkinen, K., & Laine, J. (2006). Vegetation heterogeneity and ditches create spatial variability in methane fluxes from peatlands drained for forestry. Plant and Soil, 285(1-2), 289–304. doi:10.1007/s11104-006-9016-4

10. Nichols, J. E., & Peteet, D. M. (2019). Rapid expansion of northern peatlands and doubled estimate of carbon storage. Nature Geoscience, 12(11), 917–921. doi:10.1038/s41561-019-0454-z

11. Page, S. E., & Baird, A. J. (2016). Peatlands and global change: response and resilience. Annual Review of Environment and Resources, 41(1), 35–57. doi:10.1146/annurev-environ-110615-085520

12. Page, S. E., & Hooijer, A. (2016). In the line of fire: the peatlands of Southeast Asia. Philosophical Transactions of the Royal Society B: Biological Sciences, 371(1696), 20150176. doi:10.1098/rstb.2015.0176

13. Usup, A., Hashimoto, Y., Takahashi, H., & Hayasaka, H. (2004). Combustion and thermal characteristics of peat fire in tropical peatland in Central Kalimantan, Indonesia. Tropics, 14(1), 1–19. doi:10.3759/tropics.14.1

---

## Referee Comment (RC2) · Anonymous Referee #2 · 20 Apr 2020

The authors present an interesting, mathematical optimization solution to re-wetting drained tropical peatlands. In doing this they apply an engineering approach which in itself is interesting while at the same time it may ignore some specific characteristics of tropical peatlands. Since tropical peatlands are vulnerable ecosystems the challenge is to develop a tailor made canal blocking system combining optimization techniques with good knowledge of peatland ecosystems.

In this respect comments are:

Comment 1 In practice, dimensions of drainage canals in tropical peatlands change from narrow and shallow streams close to the centre of the peat dome toward wide and deep canals at the location where the canals enter into a surrounding river. Consequently how do the authors deal with the hydrological consequences of changing

dimensions of drainage canals?

Comment 2 In relation to comment 1, water head differences over relatively simple dams in the centre of the peat dome could be substantially smaller compared to water head differences over large dams. How do authors take this into consideration?

Comment 3 Normally not the number of dams but rather the amount of money available determines rewetting activities in tropical peatlands. Since small dams are often constructed using locally available material they are much cheaper than large dams often requiring wooden foundation poles and sand bags. Consequently can the authors specify which type of dam they have in mind and what its costs are? Also is constructing smaller and therefore cheaper dams an option?

Comment 4 When blocking drainage canals it is generally advised to start placing simple dams in the upstream part of the catchment. This gradually reduces water pressure in the downstream part and thereby reduces failure of the larger dams placed in the downstream part. Do the authors envisage a similar procedure of dam construction or do they propose an alternative?

Comment 5 When constructing dams do the authors take the peat depth into consideration? Large dams generally require a foundation of wooden poles driven into the mineral subsoil and this is only feasible in shallow peat areas.

Comments 6 Do the authors recommend a cascade of several types dams along a canal ranging from simple and cheap in the upstream part and complex and expensive in the downstream part of the catchment?

---

## Author Comment (AC1) · 17 May 2020

**Canal blocking optimization in restoration of drained peatlands** (bg-2020-83)

We would like to thank the Referees for the constructive comments, which increase the quality of the paper. We have carefully considered each comment. You can find our responses below in this blue-colored font. Note that the line numbers in the comments by the referees correspond to line numbers in the old version of the manuscript, while line numbers in our responses refer to the new version.

**Referee #1**

**General comments:**

This manuscript consists of a simple, quasi-hydrologic model for water level in ditches (involving a graph problem) plus a simple hydrologic model for flow in non-ditches. These models are used like a management tool to evaluate the impact of ditches on ground water levels and greenhouse gas emissions in the peatland. The scientific question is worthy. It is not obvious that it falls within the scope of BG; however, it does involve interaction between organisms and the geosphere in the sense that organic decomposition is responsible for the emissions that are inferred from the water table estimates. It seems to me that if Hooijer et al (2010; doi:10.5194/bg-7-1505-2010) had a place in this journal, then so too does this manuscript. I have not seen such an exploration of peatland ditch blocking as an optimization problem in the literature. So the observations here, as well as some of the strategic decisions that the authors reached while designing their procedure, are novel, interesting and potentially useful. In my view, the most valuable contributions of the paper are:

1. Showing that this problem is worth solving, in the sense that computer optimization of canal block locations worked much better than manual placement based on expert rules (though these rules are not sufficiently explained; see specific comments).
2. Showing that heuristic global optimization routines, in particular, can be valuable tools in attacking practical problems of this nature.
3. Suggesting that an initial optimization step of maximizing ditch water level could provide good starting steps for a more computationally expensive analysis of water levels within peat.
4. On a more technical note, the imposition of Dirichlet conditions in the domain interior using an implicit source term is a potentially useful approach for similar problems (though also not sufficiently explained; see specific comments).

The literature review is somewhat weak, in particular with reference to the tropical peat literature, and needs to make a clearer distinction between findings from higher-latitude and tropical peatlands. However, these additional references will not greatly change the narrative.

- We have written the literature review in a more profound manner. We have added new references suggested by the referee. We have modified the Introduction to point out the differences between tropical and higher latitude peatlands. (See Specific Comments).

The results and the methods described appear valid and give no cause to suspect problems, and the simulation code has been made available (commendably). However, there is not enough detail in the manuscript to understand, even broadly, some aspects of what was done. I believe these clarifications can be made without adding supplementary material. See specific comments. The model for drydown in peat outside the canals is probably not very accurate (see specific comments), so accuracy for this specific case is somewhat questionable, but this does not affect the main contributions noted above.

- See Specific Comments.

The manuscript is well structured overall. The abstract should broadly outline the methods that were used (simulated annealing, genetic algorithms, Boussinesq, three days' drydown after initial "reset"). The Methods

and Discussion would benefit from a minor rearrangement of sections (see specific comments). The Discussion is reasonable, concise and avoids overreach.

- The abstract now contains mentions to the methods used. The Methods section was rearranged. New text was added to the Discussion section covering the helpful comments by the two referees. (See details in the Specific Comments).

There are three limitations that do not compromise the value of the manuscript but should be touched on in the Discussion and / or Introduction (for more on all of these, see specific comments):

1. Examining the area in Google Earth, it appears there are many field drains in rectangular arrays of about 60 m x 250 m that are disregarded in the simulations because of the grid resolution. Though this does not compromise the value of the paper, it does reduce the accuracy of the results and should be made clear to the reader in the Methods section and emphasized a bit more in the Discussion.
   - The referee is right: some more small field drains exist than what our dataset captures. But we are also certain that this doesn't reduce the value of the paper. On the one hand, the data we used was the best available; on the other hand, the smaller block drains are probably not worth blocking from a WTD rise optimization point of view. Nevertheless, as suggested, this point was made clear in the Study Area subsection in the Materials and Methods (P3L77), and mentioned again in the Discussion (P18L332).

2. In practice, the expected head difference across a block is an important design criterion that was not considered in the optimization.
   - The referee is right: it was not considered in the optimization. Nevertheless, we believe our approach is still useful. See Specific Comments and also Referee #2's comments and our response to them.

3. The effect of canal blocking on methane emissions should be part of an overall evaluation of impacts but relevant experimental data from tropical settings are lacking.
   - We added a small comment introducing the topic of methane emissions in the Discussion (End of "Implication to $CO_2$ emissions" section).

**Specific Comments:**

- P1 L2, Abstract: "Ecosystem restoration can be achieved by raising the water table": "Achieved" is a rather strong word; rewetting seems to be a necessary but not sufficient condition for tropical peatland ecosystem restoration.
- [corrected, P1L2]

- Introduction: unclear to a reader which results have been described in the tropics. Please distinguish references from higher-latitude peatland studies; in particular, blanket peats are rather different systems from the lowland tropical peats examined here. One approach could be to start by talking about peatlands in general, and then shift to discussing what is known from the tropics specifically.
- We have restructured the Introduction in the way the referee suggests: from general to specific issues in tropical peatlands. The references were also ordered according to this, following the referee's valuable comments.

- P1 L15: Additional references regarding peatland carbon pool: Nichols and Peteet 2019, Le Quéré et al 2018, or for a review, Page and Baird 2016.
- The suggested references were added (P1L17).

- P2 L21: For peat subsidence in the tropics, see also: Couwenberg et al 2010, Hooijer et al 2010, Carlson et al 2015.
- The suggested references were added (P2L24).

- P2 L22: Fire risk in peatlands: see also: Usup et al 2004, Page and Hooijer 2016.
- The suggested references were added (P2L25).

- P2 L25: World Resources Institute: can you find a peer-reviewed (primary literature) source that makes this or a similar point?
- The reference to World resources Institute was deleted. To make a similar point, the work by Miettinen et al. (2017) was cited. Their estimation of C emissions from tropical peatlands in Malaysia and Indonesia in 2015 corresponded to 1.6% of global fossil fuel emissions. (P2L28)

- P2 L25-26: For $CO_2$ emissions from drained peatlands in Indonesia, consider also Miettinen et al 2017.
- A reference to Miettinen et al. (2017) was introduced. (See point above).

- P2 L27: "key variable controlling $CO_2$ emissions": add "from decomposition in tropical peatlands"
- [corrected, P2L30]

- P2 L27-28: By this point, it would be less confusing to focus on tropical references for $CO_2$ emissions vs water table depth; instead of Wilson et al 2011, consider Carlson et al 2015.
- [corrected, P2L31]

- P2 L33: Similarly, instead of Päivänen and Hånell 2012, consider Jauhiainen et al 2008, Ritzema et al 2014, Dohong et al 2018.
- The reference to Päivänen and Hånell was removed (P2L36).

- P2 L35: Use tropical references again, then something like, "Studies of canal and ditch blocking in temperate peatlands have found that..."
- We added a similar sentence emphasizing that the references deal with temperate peatlands (P2L37).

- P2 L39: "This is especially important in tropical peatlands, where the canals are typically large:" Can a reference be provided? Armstrong et al (2009) describes the typical size of ditches in blanket bogs of the UK, and could be used as a point of contrast.
- Ritzema (2014) mentions canals 15 m to 20 m wide in Central Kalimantan, Indonesia. This reference was introduced. Also, Armstrong (2009) was used as a counterpoint. (P2L43)

- P2 L50: Near: "Global optimization methods are commonly used...": Provide a very brief introduction to the terminology you use from optimization theory; I would guess that a majority of Biogeochemistry readers will not be able to infer what is "global" about global optimization, what is meant by "design space", nor why it is relevant that the design space is discontinuous and non-convex. It can be short. In the same place, you should very briefly introduce simulated annealing and genetic algorithms.
- We appreciate this comment, and indeed the referee is correct here. However, we also feel that the text flow in the Introduction is currently fluent and we don't want to jeopardize that. Our workaround consisted in the following: the technical jargon about optimization algorithms was kept to a minimum in the Introduction (around P2L50). Then, in the Materials and Methods section global optimization algorithms are better introduced (as opposed to linear programming optimization problems, P10L190). Furthermore, the phrasing of that section was slightly improved to make the point of why these algorithms were chosen without delving too deep into technical descriptions.

- Section 2, Materials and Methods: I suggest starting with the site description; reasons discussed further below.

- Materials and Methods section was rearranged, and it now begins with the Study Area subsection.

- P3 L62: Can you come up with a different phrase or modifier for "the hydrological model", as the canal water level subroutine is also in some sense hydrological? Perhaps "peat hydrological model".

- All occurrences of "hydrological model" were changed to "peat hydrological model".

- P3 L66: "target variable": I believe this is the only place this phrase is used in the manuscript; consider changing to "objective function" to reduce the number of new terms for unfamiliar readers.

- We decided to keep "target variable" throughout the manuscript. However, those are not the exact same thing (i.e., the *evaluation* of the objective function gives the target variable), and the term "objective function" was used twice, always in relation to the target variable (P9L183; P10L185).

- P3 L67: "We also tested an alternative, simpler optimization approach (SO)": Simpler than what? Could be easier to follow if the SA and GA optimization approaches are introduced first.

- GA and SA are now mentioned before SO, so the "simpler" adjective makes sense now.

- P3 L73: "This subroutine calculates the CWL after building a set of blocks, v', based on their positions, k." As written, it is not clear to what v refers (a set of blocks? blocks?), nor why it carries a prime ('). I suggest something like: "This subroutine calculates the CWL v' after building a set of blocks at positions k based on the original CWL v."

- [corrected, P4L95]

- P3 L74: "the CWL is assumed to be at a fixed distance, wd, below the peat surface, s...": Here, I ask myself: is wd a product? For this reason, I would discourage using compound symbols like this, but if they are used, please clarify in some way that this is a single symbol (perhaps by referring here to the nice table of symbols).

- Compound symbol "wd" has been changed to "w"

- P3 L75, Eqn 1: Does i index over pixels? if so, how is the peat surface elevation s defined in a canal pixel? It looks like the DTM pixels are much larger than the canals are wide, so I guess that s was derived directly from the DTM elevation? This would be easier to follow if the site (and DTM) were described first.

- Eq.1 now reads $v_i = DEM_i - w$, with i ranging over canal pixels (P4L97). Also, as mentioned before, the Materials and Methods section was rearranged so that the site description is described first.

- P3 L76: "the value of wd was determined by direct observation...": Where? If at the site, it would simplify things to put the site description first. Otherwise, refer to that section.

- Yes, it was direct observation on site. Corrected (P5L99).

- P3 L77: What is the "head level" of a block?

- Block head level was explicitly defined: "The block head level is defined as the distance from the DEM elevation to the highest point of the block" (P5L108). This description was also added to the table of symbols.

- P3 L78: Change "further up the canal network" to "upstream".

- [corrected]

- P3 L77-81: Explain how "upstream" is determined prior to stating that a canal block causes the water in all upstream canal pixels to rise to the same level.
- Upstream pixels now defined in the subsection "canal water level subroutine".

- P3-4 L77-85 and Appendix A: I think "direct causal contact" or "direct physical contact" does not convey what is meant here; it would be good to find a better phrase. How about talking about "contiguous upstream pixels", and explaining that "contiguous" means not separated by a canal block?
- "Contiguous upstream pixels" was used. (See, *e.g.*, P5L103)

- P4 Figure 2: In this figure it becomes clear that v is positive up, and it appears that wd is positive down (if water table is further below surface s), but it is still not clear until Eqn 3 that WTD is positive up (even from Table 1). It would be good to mention this earlier, perhaps in Table 1, because "water table depth" causes different people to picture different things (does "a greater depth" mean the water table is higher or lower?). Throughout, I would suggest instead using "water level", after Bechtold et al (2014).
- We are aware of the typical confusion between the terms water table level and water table depth, and the related misunderstanding of the sign along the vertical direction. However, in this manuscript we would prefer to keep the word "level" as a property of the canals (hence "canal water level" or CWL), and the word "depth" as a property of the peat (hence "water table depth"). However, as suggested, we added the description "negative downwards" to the WTD in Table 1.

- P5 L90: Regarding the applicability of the Dupuit-Forchheimer assumptions: Insert "much", changing "for domains wider than they are thick" to "for domains much wider than they are thick".
- [corrected, P7L117]

- P5 Eqn 2: From Eqn 4, I believe that transmissivity T is a function of both the water table h and the elevation of the impermeable bottom ib, so if the functional dependence of T is written, it should be T(h, ib) rather than T(h).

- About writing the functional dependency of the transmissivity, T. Although we agree that strictly T(i.b., h) should be written, i.b. and h have a very different role in our model: i.b. is a (spatially varying) fixed parameter of the model, and h is the variable to be solved in eq(2). It is therefore the dependence of T on h what makes eq.(2) nonlinear, and what, in our view, needs to be emphasized. In order to clarify this point, it was explicitly addressed in the text by noting that T(ib, h) = T(h).

- P6 L96-97: How was time stepping handled? Explicit, implicit? How was convergence determined? From later in the page, it looks like time steps were fully explicit in the functions T and Sy (the value from the beginning of the time step was used)?
- The solution of Eq.(2) was fully implicit in time –both for h, and for T(h) and Sy(h). This information was added to the text (P7L126).
  About convergence: For each timestep, the amount of internal iterations to solve the discretized system of equations was set to 3. See below for a related discussion about the tradeoff between accuracy and efficiency.

- P6 L99-100: "The value of h at the canal pixels was forced to be equal to v' by adding an implicit source term large enough to completely dominate the matrix diagonal": What was done exactly?
- The method we refer to is the standard way to fix the value of the dependent variable when solving a differential equation numerically. See, e.g., Versteeg and Malalasekera, 2nd edition (2007), pages 267-268: "We need a technique to cope with situations where we need to set the value of a variable at a node. This can be done by introducing two overwhelmingly large source terms into the relevant

discretized equation." See also the example provided therein. Regarding the manuscript, the sentence was made more accurate and the reference to Versteeg and Malalasekera was added (P7L130).

- P6 L110-114: Move the sentence "The van Genuchten function was used..." to before the sentence "In absence of measured..." (assuming that data from Päivänen 1973 were used to parameterize the van Genuchten function?)
- [corrected, P8L140]

- P6 L110-114: Plot the resulting specific yield and transmissivity functions. Transmissivity could be plotted for the lowest substrate elevation, for example (or curves with different substrate elevations could be plotted together).
- Transmissivity and specific yield functions were plotted in Figure5.

- P6 L115: Were the values of T and Sy from the beginning of the time step used during time stepping? In any case, depending on the transmissivity function, I would guess the time discretization error with a daily time step could be substantial. But, the error could be acceptable as a tradeoff against runtime (at least when finding good candidate block positions). Convergence could be tested via multiple runs with decreasing time steps, but in my view is not strictly necessary for this paper.
- We added the suggested details about the numerical schemes to the manuscript: The solution of Eq.(2) was fully implicit in time –both for h, and for T(h) and Sy(h). This information was added to the text. About convergence: As explained above, or each timestep, the amount of internal iterations to solve the discretized system of equations was set to 3. We agree that in standard applications of such a hydrological model, convergence of the numerical scheme should be more thoroughly studied. As the referee points out, however, in the task of finding optimal block positions, the tradeoff with runtime is a very sensitive issue. The level of precision achieved with 3 internal iterations per timestep was judged to be an acceptable solution of the PDE. These ideas were added to the manuscript both in the Material and Methods section and in the Discussion.

- P6 L118-120: The broad outline of the simulation scenario (3 days of drydown from an initial "reset") are an important part of (SA and GA) objective function evaluation and should appear in the Abstract and the end of the Introduction.
- One-line description of methods used was added to the Abstract (P1L8) and the Introduction(P2L54).

- P7 L128: Does the spatial average of water table depth include canal pixels?
- Yes, the spatial average of Eq.(5) includes canal pixels. It was made explicit in the text (P8L158).

- P7 L128-133: I suggest dropping the subscript for the number of days averaged; it does not seem important for explaining the results and removing it would allow remov ing an equation (7).
- The 3 day drydown is a feature of our work that we believe should be emphasized. That is why we decided to make an explicit statement (Eq.7) about the use of the averaged WTD without any subscripts. Also, as a positive side effect, it simplifies the text around Eq.(8), where we would have had to introduce an extra symbol to account for the yearly averaged WTD. For these reasons, we prefer to keep the subscript around Eq.(7).

- P7 Eqn 8: I am confused about sign issues in this equation (and / or Eqn 9). If I understand correctly, zeta is positive up, so this equation seems to suggest that emissions are larger if the mean water table is higher, which is surely not what is intended.
- The sign of Eq.(8) was corrected.

- Eqn 9: Are these parameter values, as well as the affine assumption (Eqn 8), from Jauhiainen et al 2012? If so, add a citation immediately after Eqn 9.
- [corrected, P9169]

- P8 L149-150: "Let k = (k1, ..., kn) be the vector of block positions": How about: "Let k = (k1, ..., kn) be the boolean vector indicating presence or absence of a block in each canal pixel..."
- [corrected, P9L180]

- P8 L158-159: "design space", "search space": pick one, and use it also in the Introduction (P2 near L50).
- The term "search space" was selected and "design space" is not used in the text.

- P8 L164: "Genetic algorithm... and simulated annealing... can find the global minimum with high probability": true for some problems, but it is not hard to cook one up for which they would not have odds that are good at all; for example, imagine an objective function that is 0 at a single point in the plane and 1 everywhere else. How about, "... can often find the global minimum in many problems."
- [corrected, P10L196]

- P9 Table 2: The empty column under "rule-based" is confusing. How about putting something here, like "Manual", or otherwise removing the column and simply describing it in the table caption.
- The word "manual" was added to describe the rule-based approach in Table 2.

- P9 L195-196: Regarding the rule-based configuration: What were the rules?
- The first recommendation of Ritzema *et al.* (2014) for designing blocking strategies is to build blocks in "canals running perpendicular to the contour lines of the peat dome and connecting the rivers". The rule of the rule-based approach was exactly that: build blocks in perpendicular intersections of contour lines and the canal network. The text now explicitly mentions this (P11L227). The reference to Armstrong *et al.* (2009) was replaced by a reference to Ritzema *et al.* (2014), which is what was intended in the first place.

- P10 Eqns 15 and 16: It is not clear to me why an absolute value was used here. Why not just order the operands so that improvement in the mean WTD (higher water level) results in a positive value? Surely, if an intervention were to somehow lower the water table, this should result in a negative value (even if none of these interventions did so).
- Eqs. 15 and 16: the absolute value was removed, and the terms rearranged.

- P10, section 2.2, Study area: How was the DEM derived?

- The DEM source was referenced (Vernimmen, 2019). (P3L81)

- P12 Figure 5: Nice figure. I guess the multiple water level lines are for the 3 consecutive days of drydown?

- Caption of Figure 5 was modified by mentioning that the blue lines are the WTD at 3 consecutive drydown days.

- P13 L248-249: "An informative way to gauge this difference...": where can I see this in the data?
- A reference to Figure 7 (b) was added (P13L264).

- P14 Figure 7 caption: What is meant by "The random range was linearly interpolated"?

• This line intended to make the following point clear: the x axis in the figure is integer-valued, and therefore, *sensu stricto,* there cannot exist a continuous range as the one shown in Fig7. But this is self-evident, and this line makes the simple message difficult to interpret. Therefore, it was removed from the text.

- P15 Figure 9: Improve caption; not obvious what is being shown without reference to the text.

• Caption of Figure 9 was made more explicit.

- P16 L287: "this work introduces the first systematic tool..." It's hard to be sure; there could easily be something like this in use by large private landowners. How about "the first freely available tool" or "the first published study"?

• [corrected, P16L303]

- P17 L301: "the three-day average of the WTD would 2.13% units lower": I don't follow; where is this shown?

• This sentence was removed from the manuscript. It was intended to be an example of different outcome for a different parameterization. However, the referee was rightfully confused, since the sentence was oddly placed.

- P17 L306-: "Some remarks about the assumptions...": This would be a good place to mention any other limitations of the DEM. How was it derived? Do you believe it to be highly accurate?

• We used the best available DEM, described in Vernimmen et al. (2019). The DEM was preprocessed with the *fill sinks* algorithm in QGIS 3.4 in order to indentify and fill unwanted surface depressions. This information was introduced in the Materials and Methods section. However, no such elevation model is perfect and we are aware that it may contain inaccuracies that can slightly contribute to errors in the average WTD.

- P17 Section 4.2: This section starts by talking about the rule-based method, then discusses the optimization results, and then comes back to compare the optimization results to the rule-based method. This section could be made easier to follow, and some redundancy might be eliminated, by starting with the optimization results and then contrasting the results from manual (and random) block placement.

• We decided to keep the structure in this section. Although the referee is right in the comment that the text feels slightly redundant with this structure, the idea behind it is to emphasize the idea that placing the blocks randomly is similar to doing so by rule-guided human guesses. This is a line of reasoning that we think some readers may criticize (although we think is very solid), and we want to make it as explicit as possible.

- P17 L322: "The positions for the blocks in the rule-based approach were based on the contour map..." Derived how? Besides explaining this in the Methods, it would be good to briefly mention the criteria for manual block placement here.

• [corrected, P18L343]

- P19 L370-379: The discussion of why the idea of steady-state Boussinesq solution was rejected could be condensed.

• The discussion was condensed (starting P19L393).

- P19 Section 4.3: In this section, it would be good to at least briefly mention the possible effects of canal blocks on methane emissions from ditches. From higher-latitude peatlands there are a number of studies, in Finland and elsewhere, e.g., Minkkinen and Laine (2006). In the tropics much less work has been done; without doing a search, I am aware of these two: Jauhiainen and Silvennoinen (2012), Manning et al (2019).

- A small mention to the possibility of accounting for methane emissions was added to the Discussion (starting at P20L416), and some of the suggested references were used. Given that methane emissions seem to increase with rising WTD, it opens up the possibility for a very interesting question from the optimization point of view: the approach should not focus in maximizing WTD rise, but rather in balancing it so as to minimize total C emissions.

- P19 L384-385: 80 blocks results in about 14 km between blocks: you could state more explicitly that your method remains applicable for placement of a larger number of blocks (at the expense of more computing time). This could be a good place to mention the typical design criterion, not considered in this study, of avoiding large head differences across blocks to prevent dam failure (per general comments).
- We stated more explicitly that our method remains applicable for placement of a larger number of blocks (P20L404). The comment about the large head difference is similar to the comments of Referee #2. See response to Referee #2 for more details. Also, this issue is included in the manuscript in a newly created subsection of the Discussion: Application to real-life scenarios (starting at P20L421).

- P19 L392-394: Good point but phrasing could be improved.
- Phrasing was improved (P20L413)

- P20 Algorithm 1: v'A <- vA + hl: Appears not to match Figure 2, or Table 1; looks like it should be v'A <- sA - hl.
- The referee is right. It wa corrected (P22, Algorithm 1)

**Technical corrections:**
- P1 L4: Change "water level raise" to "water level rise"
- All errors regarding "raise" vs "rise" were corrected.

- P3 L77: Change "its water level raises up" to "its water level rises up".
- [corrected]

- P4 Figure 1: The references to text sections look wrong (assuming that is what they are?): I guess 2.2 in "Canal water level subroutine (2.2)" should be changed to 2.1.1, etc.
- [corrected]

- P8 L170: Change "the only parameter" to "the sole parameter".
- [corrected, P10L202]

- P8 L172: Add quotes around "individual", because it is being used in the GA sense of an "organism". Change 'individual k' to '"individual" vectors k'.
- The word *individual* was italized (P10L204).

- P8 L174: "our implementation": the analysis used an existing implementation of the algorithm, no? Perhaps, "... were the only parameters in the genetic algorithm implementation we used."
- [corrected, P10L206]

- P9 L177: Change "wide spread classical single processor algorithm" to "single processor algorithm".
- [corrected, P10L209]

- P9 L188: Change "over 10 processors" to "on 10 processors".
- [corrected, P10L210]

- P10 L222: Change "sappric" to "sapric".
- [corrected, P8L137]

- P11 Figure 4: Colors for "2 m" and "8 m" bins are hard to distinguish; why not use another continuous scale?
- The contrast in the colours of both the peat type and peat depth maps was increased. Due to rearrangement Figure 4 is now Figure 2.

- P13 L262: Remove "eloquently".
- [corrected, P14L276]

- P14 Figure 7: What are the plectrum-like markers?
- See next comment.

- P14 Figure 7: How do I find the "natural" and "drained" areas discussed later in the text?
- The plectrum-like markers show the natural and drained areas discussed later in the text. The caption of Fig7 was made more explicit to this respect.

- P14 Figure 7: In the plots, make the markers bigger. They are hard to distinguish, and it is hard to make out their colours and shapes.
- We did not change the markers, since it is our understanding that the size of the figures in the edited version of the manuscript will be larger.

- P15 Figure 8: Move labels away from markers to make them easier to read.
- This turned out to be a surprisingly difficult task with our plotting software. We think the current position of the label markers is good enough to understand the content, even if the outcome is not as aesthetically appealing as it could be.

- P15 Figure 8: Make it more obvious to the reader what is better and what is worse on the vertical axis.
- A single sentence was introduced to explain the interpretation of the vertical axis.

- P15 Figure 8: CWL change (m) ranges to 1000; unit error? Or is this a sum?
- There is no unit error there. CWL change is indeed the sum displayed in equation 13.

- P16 Figure 10: Text for legend is very small; move outside axes? Or, consider labeling the lines directly; it takes some work, with reference to the caption, to figure out which line is which.
- [corrected]

- P16 292: Change "lowers" to "decreases".
- [corrected, P17L309]

- P18 L340: Change "any" to "every"

- [corrected, P19L362]

- P20 L418: Change lh to hl.

- [corrected]

**Referee #2**

**Referee #2 comments:**

The authors present an interesting, mathematical optimization solution to rewetting drained tropical peatlands. In doing this they apply an engineering approach which in itself is interesting while at the same time it may ignore some specific characteristics of tropical peatlands. Since tropical peatlands are vulnerable ecosystems the challenge is to develop a tailor made canal blocking system combining optimization techniques with good knowledge of peatland ecosystems.
In this respect comments are:

**Comment 1** In practice, dimensions of drainage canals in tropical peatlands change from narrow and shallow streams close to the centre of the peat dome toward wide and deep canals at the location where the canals enter into a surrounding river. Consequently how do the authors deal with the hydrological consequences of changing dimensions of drainage canals?

- There was not enough data to assess canal width in a quantitative way. Moreover, if it had existed, it would have been restricted by the resolution of the underlying raster maps. In our case all the raster maps (DEM, peat depth, peat type) had a 100m x 100m resolution. If information about canal dimensions would have been available, one possibility would have been to resize the rasters to a finer grid resolution in order to be able to capture the canal width explicitly. However, we believe that doing so would slow the computation of the groundwater table to the point that no numerical optimization could have had taken place. Therefore, varying canal width was not accounted for.

**Comment 2** In relation to comment 1, water head differences over relatively simple dams in the centre of the peat dome could be substantially smaller compared to water head differences over large dams. How do authors take this into consideration?

**Comment 3** Normally not the number of dams but rather the amount of money available determines rewetting activities in tropical peatlands. Since small dams are often constructed using locally available material they are much cheaper than large dams often requiring wooden foundation poles and sand bags. Consequently can the authors specify which type of dam they have in mind and what its costs are? Also is constructing smaller and therefore cheaper dams an option?

**Comment 4** When blocking drainage canals it is generally advised to start placing simple dams in the upstream part of the catchment. This gradually reduces water pressure in the downstream part and thereby reduces failure of the larger dams placed in the downstream part. Do the authors envisage a similar procedure of dam construction or do they propose an alternative?

**Comment 5** When constructing dams do the authors take the peat depth into consideration? Large dams generally require a foundation of wooden poles driven into the mineral subsoil and this is only feasible in shallow peat areas.

- Response to comments 2, 3 and 5: These comments contribute substantially to the potential impact of our approach to real-life scenarios. One of the underlying assumptions of our method is that blocks can be built at any point in the canal network, and that the cost of doing so is constant and independent of any other variables. This is, of course, just a rough approximation. The referee's comments are very valuable in identifying the specific variables that should be included in a better model of block cost. The optimization scheme presented in the manuscript can take these factors into account rather straightforwardly: peat depth and site topography are already part of the input data, and a function that computes the cost of a block depending on its dimensions and location could replace our current constraint in the optimization problem. This kind of work, however, calls for onsite collaborations with

local organizations. All these ideas were included in a newly created subsection of the Discussion: Application to real-life scenarios.

**Comment 6** Do the authors recommend a cascade of several types dams along a canal ranging from simple and cheap in the upstream part and complex and expensive in the downstream part of the catchment?

- Response to Comments 4 and 6: Possible failure of dams was not considered in our study. Neither was the fact, as mentioned before, that different types of blocks should be considered when blocking points of the canal network with different properties. Armstrong et al. (2009) made a comprehensive study of several drain blocking strategies in UK blanket peatlands and proposed a best-practice-guide decision tree based on it. His analysis considers variables such as the slope of the gradient, peat wetness and walking distance to construction sites in a qualitative way. One way of including these variables in our scheme would be to modify the aforementioned cost function of the blocks: a block that is more likely to fail, one that requires a greater structure (such as bypasses), or one that is located in a remote spot, is a more expensive block. Our proposal for improving block locations does not replace this type of expert knowledge, but rather it should build upon it in order to have the desired practical impact. On the other hand, it remains true that choosing the location of a set of blocks for best performance is a daunting task due to the complexity of the response of the water table. Our approach tries to facilitate that task by acknowledging that expert knowledge alone might not be enough to solve the problem, and opens up the opportunity to work together with process-based hydrological modelling and numerical optimization techniques, which turn out to be powerful tools. All these ideas were included in a newly created subsection of the Discussion: Application to real-life scenarios.

**REFERENCES**

Armstrong, A., Holden, J., Kay, P., Foulger, M., Gledhill, S., Mcdonald, A. T., and Walker, A.: Drain-blocking techniques on blanket peat: A framework for best practice, Journal of Environmental Management, 90, 3512–3519, https://doi.org/10.1016/j.jenvman.2009.06.003, 2009

Miettinen, J., Hooijer, A., Vernimmen, R., Liew, S. C., & Page, S. E. (2017). From carbon sink to carbon source: extensive peat oxidation in insular Southeast Asia since 1990. Environmental Research Letters, 12(2), 024014. doi:10.1088/1748-9326/aa5b6f

Ritzema, H., Limin, S., Kusin, K., Jauhiainen, J., and Wösten, H.: Canal blocking strategies for hydrological restoration of degraded tropical peatlands in Central Kalimantan, Indonesia, Catena, 114, 11–20, 2014.

Vernimmen, Ronald; Hooijer, Aljosja; Yuherdha, Angga T.; Visser, Martijn; Pronk, Maarten; Eilander, Dirk; Akmalia, Rizka; Fitranatanegara, Natan; Mulyadi, Dedi; Andreas, Heri; Ouellette, James; Hadley, Warwick: "Creating a Lowland and Peatland Landscape Digital Terrain Model (DTM) from Interpolated Partial Coverage LiDAR Data for Central Kalimantan and East Sumatra, Indonesia." *Remote Sens.* 11, no. 10: 1152, 2019.

Versteeg, H. K. and Malalasekera, W.: An Introduction to Computational Fluid Dynamics -The Finite Volume method, Pearson Education Limited, Second edition, 2007.

---

## Author Comment (AC2) · 29 May 2020

Dear Referee#1,

You may find the response to your deep and thoughtful comments in the separate Author's Comment entry.

By means of this note I would like to amend one statement made in there. About the comment P6 L96-97, I wrote that the solution was implicit for both h and S(h) and T(h). However, it was only implicit for h, and explicit for S(h) and T(h): as you pointed out in your comment, the value of the beginning of the time step was used to solve the discretized equation.

Best regards,

Iñaki Urzainki (main author)

---

## Author Response (AR2)

**Response to second round of referee comments**

1. "Drained peatlands are one of the main sources of carbon dioxide (CO2) emissions globally."
- Could this statement in the abstract be supported with a reference in the text, perhaps by adding a clause to the sentence, "However, drainage may turn peatlands into C sources..." (lines 19-20)?

We opted for the following construction:

"However, drainage may turn peatlands into C sources (Minkkinen and Laine,1998; Hooijer et al., 2010; Ojanen et al., 2010; Jauhiainen et al., 2012), and as a consequence drained peatlands are one of the main sources of CO\textsubscript{2} emissions globally."

2. Lines 11-12: "Our solution consistently improved the performance of traditional block locating methods"
- Consider "Our solution consistently outperformed manual block locating methods..."

corrected

3. Line 41: "[T]he number of blocks becomes easily limited by available resources"
- "Available resources" is slightly confusing here because the paragraph begins by discussing the material used for blocking. I had the image of running out of peat for the blocks. Change perhaps to "available financial resources", or just "cost"?

We opted for "available financial resources".

4. Line 42: "requiring big constructions"
-> "requiring large structures"

corrected

5. Line 43: "amount"
-> "number" of blocks.

corrected

6. Line 84: "and the peat depth defines the impermeable bottom ib"
- Perhaps "the peat thickness defines the depth ib of the impermeable bottom below the peat surface"? I found myself wondering, at this point in the text, what exactly ib was (the elevation of the bottom? Is it a scalar or a field?)

Corrected. It is a field, and it should be clear from the reference to Figure 2(c) and from the discussion below equation (4).

7. Line 88: "[The peat hydrologic model] solves the WTD for the whole area, a quantity that is closely related to the target variable... ":
- Consider rearranging to: "... solves for WTD in the whole area. WTD is closely related to the target variable... "

corrected

8. Line 90: "This onsets a new iteration."
-> "This starts a new iteration."

corrected

9. Fig 3: Blue text is much smaller than the text in boxes, and thus a little hard to read. Blue text could be made bigger, or the text in boxes smaller.

Corrected

10. Line 108: "instead of using the block height as a variable we use its complementary, the block head level hl."
- Delete "its complementary,"

corrected

11. Lines 126-127: "The numerical scheme was fully implicit in time for h, and explicit for T (h) and S y (h)."
- The response to reviewers says: "The solution of Eq.(2) was fully implicit in time –both for h, and for T(h) and Sy(h)." Which was it?

The one in the final text is right: the scheme was implicit in time for h, but explicit for T(h) and Sy(h). The editor was already informed by email about this error in the response to reviewers.

12. Line 131: T(h) = ...
- At this point I wonder what exactly ib is (forgetting whatever I learned from previously reading the manuscript). I think to myself: In the equation, I am integrating from ib to h, so I guess z is defined relative to the peat surface, and ib is the depth of the impermeable bottom below the surface? Is ib a scalar or a field? In line 133, I read "ib has a known, fixed value for all the domain", which makes me think it is a scalar. Some minor editing here could make this easier to understand.

This issue was resolved in the following way: first, the definition of ib is improved: "ib is the depth of the impermeable bottom relative to the peat surface". Second, instead of "ib has a known, fixed value", it now reads: "since ib is directly inferred from peat depth measurements (see Figure 2)". It should now be clear what ib stands for and that it has a known, fixed but spatially varying value. Also, to stress that point, the dependence of ib on x and y is explicit in equation (4).

13. Line 191: "Therefore, this problem is better tractable with non-linear, global optimization algorithms."
- Consider changing "tractable" to "addressed"

corrected

14. Line 193: "Even global optimization algorithms are not guaranteed to find the optimal solution in a convex search space..."
- Delete "convex" (the same is certainly true of a non-convex search space).

corrected

15. Line 201: "In SA this is achieved by allowing disimprovements with certain probability."
- Consider: "... this is achieved by allowing steps that worsen the objective function with certain probability."

corrected

16. Line 219: "would need"
-> "requires"

corrected

17. Line 227, explanation of criteria for manual block placement:
- This is great; clear, and fully resolves my comment.

18. Fig 5 caption "a straight horizontal line indicating the cross-section area"
-> "... indicating the location of the cross-section shown in (b)."

Corrected. (We included also sub-figure (c)).

19. Fig 5 caption: "amplified"
-> "magnified"

corrected

20. Fig 6 caption: "... their effectiveness varies from each case depending on the local topography..."
- Delete "from each case"

corrected

21. Fig 7:
- Figure looks much better; great.

22. Lines 332, 463, and 468: "causal contact"
- Change to phrasing using "upstream" and "contiguous" for consistency.

corrected

23. Line 336: "... in reality w is not constant but it might vary in time due to seasonality"
- Define with a phrase, so your reader doesn't have to consult Table 1; something like: "... in reality the initial water table depth w..."

corrected

24. Line 364: "As the number of blocks to locate, b, increases, the size of the search space does so as [n choose b]. It is this exponential increase in computational complexity..."

- I suggest deleting "exponential"; the size of n choose b has a maximum near b = n / 2, so it might be better if the reader doesn't think about it too much. Otherwise, perhaps something like "rapid increase in complexity for reasonable numbers of blocks..." or something like that.

corrected

25. Page 19: There is some repetition here; could text be condensed?

The referee is right; the text was repetitive in the paragraphs about the relative improvement and the marginal benefit, MB. Since the information those metrics deliver is partially overlapping, those two paragraphs were merged into a single one (now extending from P19, lines 372 to 387), and the repetition reduced.

26. Line 396: "In tropical climates... rainfall intensity is all but uniform in time."
- Try a different phrasing here; the "all but" English idiom has a meaning that is the opposite of what is intended here (https://english.stackexchange.com/questions/9967/all-but-idiom-has-two-meanings)
Perhaps: "In tropical climates... rainfall intensity is highly variable in time". The phrase "... rainfall intensity is anything but uniform in time" has the desired meaning but is colloquial.

Thanks for the information! Corrected

27. Line 415: "To get a grasp of the magnitude of these numbers, they are of the order of what 25000 cars with an annual mileage of 20000 kilometres would emit."
- This is great. Is there a reference (possibly a website on fuel emissions) that could be cited here?

Unfortunately, we could not find any citable sources that would support this claim. However, it is a very naïve estimate with the only purpose of helping to bring large numbers closer to common sense. We think that the referees would agree in that it isn't relevant for the scientific value of the manuscript, and thus we would be glad to keep it even though we could not find such a reference.

28. Line 416: Brief section regarding methane
- Great.

29. Line 434: "Regarding the formulation of the optimization problem, block cost could be introduced by changing the constraint equation"
- Could briefly mention here, if you want, that a convenience of "black box" optimizers like SA and GA is that it is trivial to accommodate a more complex objective function.

We added the world "simply", so that the same idea is captured without going into too much detail. The line now reads: "
[revised manuscript text omitted]